# Determinants of regulatory compliance in health and social care services: A systematic review using the Consolidated Framework for Implementation Research

Paul Dunbar[1]*, Laura M. Keyes[1], John P. Browne[2]

1 Health Information and Quality Authority, Mahon, Cork, Ireland, 2 School of Public Health, University College Cork, Cork, Ireland

* pdunbar@hiqa.ie

## Abstract

### Background

The delivery of high quality care is a fundamental goal for health systems worldwide. One policy tool to ensure quality is the regulation of services by an independent public authority. This systematic review seeks to identify determinants of compliance with such regulation in health and social care services.

### Methods

Searches were carried out on five electronic databases and grey literature sources. Quantitative, qualitative and mixed methods studies were eligible for inclusion. Titles and abstracts were screened by two reviewers independently. Determinants were identified from the included studies, extracted and allocated to constructs in the Consolidated Framework for Implementation Research (CFIR). The quality of included studies was appraised by two reviewers independently. The results were synthesised in a narrative review using the constructs of the CFIR as grouping themes.

### Results

The search yielded 7,500 articles for screening, of which 157 were included. Most studies were quantitative designs in nursing home settings and were conducted in the United States. Determinants were largely structural in nature and allocated most frequently to the inner and outer setting domains of the CFIR. The following structural characteristics and compliance were found to be positively associated: smaller facilities (measured by bed capacity); higher nurse-staffing levels; and lower staff turnover. A facility's geographic location and compliance was also associated. It was difficult to make findings in respect of process determinants as qualitative studies were sparse, limiting investigation of the processes underlying regulatory compliance.

**Data Availability Statement:** All relevant data are within the manuscript and its Supporting Information files.

**Funding:** This review comprises part of a PhD study funded by the Health Information and Quality Authority (Ireland). This research was conducted as part of the Structured Population health, Policy and Health-services Research Education (SPHeRE) programme (Grant No. SPHeRE/2019/1). The funders had no role in study design, data collection and analysis, decision to publish, or preparation of the manuscript.

**Competing interests:** PD and LMK are currently employed by the Health Information and Quality Authority, which is the regulatory authority for health and social care in Ireland. However, this does not present any conflict of interest for this review.

## Conclusion

The literature in this field has focused to date on structural attributes of compliant providers, perhaps because these are easier to measure, and has neglected more complex processes around the implementation of regulatory standards. A number of gaps, particularly in terms of qualitative work, are evident in the literature and further research in this area is needed to provide a clearer picture.

## Introduction

The delivery of high quality care is a fundamental goal for health systems worldwide. Quality is variable, due to structural issues such as insufficient staffing levels [1], or process issues like poor cleaning practices [2], and can cause differences in outcome across health and social care providers such as high complication rates and poor patient experience [3].

Governments commonly introduce regulation to monitor the quality of goods and services. Regulated organisations/individuals must comply with prescribed standards. In health and social care, regulatory bodies typically license or authorise providers, and/or directly regulate the structures and processes of care through inspection, feedback or sanctions [4].

Regulation can be defined as: "sustained and focused control exercised by a public agency over activities which are valued by a community" [5]. The scope of regulation in health and social care differs depending on the setting and country. Principally, regulation centres on structural, process and outcome-related aspects of services. For example, in nursing homes (NH) in the United States, regulations cover *inter alia* resident rights, nursing services, infection control, physical environment and resident assessment [6].

There are important differences in how countries regulate health and social care. In one model the regulator is a publicly-funded, independent organisation which is aligned to a government ministry e.g. Healthcare Improvement Scotland [7]. There are also examples of regulatory authorities that are incorporated into government ministries e.g. Ministry of Long-Term Care (Ontario) [8].

In both examples above, the regulator has statutory powers of inspection and enforcement. A third approach is found in countries such as Australia where mandatory accreditation is performed by third-party organisations [9]. While having no statutory powers their role is akin to a regulator because any decision to not accredit can result in State sanctions [10]. This is distinct from other models of voluntary accreditation [11].

Regulatory compliance can be understood as "behavior fitting expectations communicated to regulatees regarding how the former should or should not behave in a given domain" [12]. This conceptualisation of compliance is markedly different to adhering to clinical guidelines or voluntary codes of practice. A feature of regulatory compliance is the mandatory requirement to comply with standards and rectify deficits.

There is a substantial literature on the structures (e.g. public/private ownership, facility size and staffing levels/competencies [13–16]) and processes (e.g. disposition towards regulation, normalisation of compliance within day-to-day operations [17, 18]) which determine compliance by health and social care organisations. This literature, not yet synthesised, is a distinct subset of the broader literature on determinants of quality because of the context and focus of regulation. Unlike most quality assessment initiatives, regulation involves consequences for the regulatee, and therefore introduces different motivations around implementation.

Regulators and regulatees would prefer that all interactions produced findings of compliance. Understanding the determinants of compliance is one way to increase this likelihood. Regulation can be regarded as a complex intervention and compliance equates to successful implementation. This allows for the use of implementation science concepts to better understand why some organisations fail to comply: "implementation science enables questions to be asked about whether, and if so how, an intervention can make a difference to a patient's life or to the practice of a health care delivery team" [19]. The success of an innovation is contingent on a range of factors. For example, the attitudes and beliefs of staff towards an innovation can influence its implementation [20].

Several tools/frameworks have been developed in the field of implementation science. The Consolidated Framework for Implementation Research (CFIR) is one such framework [21]. The CFIR was developed by drawing together disparate existing theories on implementation to produce an "overarching typology–a list of constructs to promote theory development and verification about what works where and why across multiple contexts" [21]. The CFIR has been used previously as a framework for structuring and synthesising the results of systematic reviews [22].

There is a body of work critiquing the effectiveness of regulation (e.g. improving system performance [23]), but given that regulation exists, and that there is much effort put into compliance, it is legitimate to investigate this practice on its own terms. There has been no previous synthesis of the extensive literature which makes use of regulatory compliance as an outcome measure in health and social care settings. Moreover, little is known about the processes by which health and social care services manage regulatory encounters and how this impacts on compliance. Thus, the aim of this systematic review is to identify and describe determinants of regulatory compliance in health and social care services, and the broader phenomenon of attempting compliance. The focus of the review is on the regulation of organisations as there are existing evidence syntheses on the regulation of individual professionals [24, 25].

## Methods

We performed a systematic review of studies that used qualitative, quantitative or mixed-methods approaches to identify determinants of compliance in health and social care services. We take regulation to mean any form of evaluation of quality and safety which is underpinned by sanctioning powers. Any literature focusing on the regulation of individual professionals (e.g. nurses) or on occupational health and safety (e.g. working hours) were outside scope. Determinants were anything associated with compliance and could be aspects of the organisation or the environment in which it is situated. Determinants may also be barriers or facilitators to successful implementation of regulatory requirements. In quantitative studies the outcome of interest was regulatory compliance and in qualitative studies this was also the phenomenon of interest. We interpreted regulatory compliance as any formal approval of the performance of a health or social care provider by the relevant regulator or accrediting agency. The review follows a published protocol which was not registered [26]. The methods applied are summarised in brief with additions to/deviations from the original protocol outlined.

### Criteria for inclusion

The phenomena of interest were determinants of regulatory compliance in health and social care services.

Articles—either qualitative, quantitative or mixed-methods—were included if they:

- Described factors or characteristics that were related to regulatory compliance. Specifically, this refers to regulations that are mandated by government or other state authorities. A wide range of constructs were considered for inclusion including, but not limited to, the following: service characteristics (size, location, model of care, ownership); organisational characteristics (culture, management/governance structure, maturity); service user characteristics (age, disability type, disease/illness); nature of regulatory engagement (punitive, adversarial, collaborative).

- Discussed barriers or facilitators to regulatory compliance for health and social care services.

- Were focused on quality of care in health and social care services and used regulatory compliance as an outcome measure.

  Studies were excluded if they:

- Analysed regulatory compliance in a field other than in a health or social care setting.

- Analysed compliance with clinical guidelines or other evidence-based methods for managing care that were not underpinned by the potential for regulatory sanction where there was a failure to comply.

- Used an outcome measure that was not equivalent to regulatory compliance in accordance with the definitions set out above. For example: adherence to voluntary standards or codes of conduct; where failure to comply does not result in regulatory sanctions of enforcement; compliance concerning individuals as opposed to organisations as is the case with regulations for specific health care professionals.

A search strategy using the CIMO (Context, Intervention, Mechanism, Outcome) framework was devised (S1 File). Searches were carried out on 22$^{nd}$ July 2022 on PubMed, MEDLINE, PsycINFO, CINAHL, SocINDEX and OpenGrey. Targeted web searches were conducted. The reference lists of any systematic reviews identified were searched for relevant material. In deviating from the protocol, we included the term 'accreditation' in the search terms to identify studies where accreditation was mandatory i.e. *de facto* regulation. There were no time or language restrictions. The reference lists of all included articles were hand-searched for additional relevant material. Forward citation searching of included articles was also conducted.

### Screening and full-text review

All articles were imported into Covidence [27] and the title/abstract screened by two reviewers independently. Full-text review was carried out by PD.

### Data extraction

Data extraction was performed by PD. LMK independently performed data extraction on 10% of articles for cross-check purposes. For all studies, the following data were extracted: full reference, study type, publication type, country, study aim(s), population/setting. The extraction of results data differed depending on the study design. For example, in studies reporting quantitative results, the variable name and its direction of association with compliance were extracted whereas for qualitative studies whole sentences or phrases were extracted.

Determinants expressed as composites (e.g. Medicaid occupancy multiplied by Medicaid rate) were omitted. We focused on determinants of overall compliance measures. Where not available, determinants of compliance with specific regulations were reported. We excluded variables acting as measures of quality (e.g. proportion of patients/residents with pressure

sores; proportion of patients/residents subject to physical restraint), because of the likelihood of circular reasoning: in theory, quality is an outcome, rather than a determinant of compliance.

In qualitative and mixed-methods studies, anything mentioned as influential with respect to compliance activity, either negative or positive, was extracted for synthesis.

## Analysis

We counted studies where a determinant was estimated and recorded the association with compliance in each case. In quantitative studies associations with compliance were recorded as 'positive/negative/null' for continuous or ordinal variables, 'different/null' for nominal variables ('different' denoting that one or more categories in the variable was positively or negatively associated with compliance). In qualitative studies the role of the relevant factor in influencing engagement with the regulatory process was noted.

Determinants were then coded to the most appropriate CFIR construct through an iterative and deliberative process among all three authors. PD first coded each determinant to the most appropriate CFIR construct (e.g. determinants related to facility size were coded to the structural characteristics construct in the inner setting domain), with reference to established definitions [21]. These data were then discussed jointly by PD and LMK in order to ensure that determinants were placed in the most appropriate construct. This process resulted in some determinants being moved to alternative constructs and also the creation of a category of 'other' within each domain. This was necessary due to the significant number of determinants that had no appropriate match in the CFIR (e.g. the survey tools used by inspectors had no equivalent match for a construct in the intervention characteristics domain).

PD, LMK and JPB subsequently discussed all coding with a particular emphasis on whether the determinant had relevance in the specific context of implementing regulations. For example, a determinant relating to the level of public/state funding made available for services was originally coded to the available resources construct in the inner setting domain. On reflection, we determined that this was not appropriate as the funding was not specifically related to implementing regulations. As such, this determinant was moved to 'other' in the inner setting domain. This process continued until all three authors agreed on the coding of each determinant. The analysis was also consistent with the application of the CFIR to post-implementation evaluation of innovations, where determinants are linked to outcomes i.e. compliance [21].

## Quality appraisal

Quality appraisal of quantitative and qualitative studies used the Joanna Briggs Institute tools [28]; for mixed-methods studies we used the Mixed-Methods Appraisal Tool (MMAT) [29]. All grey literature were PhD theses and were appraised using the most appropriate Joanna Briggs Institute tool [28]. PD and FB (see acknowledgments) carried out quality appraisal independently and subsequently compared findings. Any disagreements were resolved by PD and FB jointly reviewing the respective studies and arriving at a consensus.

## Results

7,500 records were retrieved: 7,383 from electronic database searches and 117 from targeted web searches, reference searches and forward citation searching. Of the 7,383, 3,739 duplicates were removed. Title/abstract of 3,644 were screened, 3,443 did not meet inclusion criteria, leaving 201 for full-text review. All but two studies were retrieved, leaving 199 for full-text review. Upon full-text review, 98 were excluded leaving 101.

Of the 117 found by other means (i.e. targeted web searches, reference searches and forward citation searching), 61 were ineligible, leaving 56. The total number of studies in the review was 157 (See Fig 1 for PRISMA flow diagram, S1 Checklist for PRISMA checklist and S2 Table for full list of included studies). Characteristics of included studies are set out in Table 1.

Search parameters were configured to identify studies that specifically focused on compliance as an outcome. It is possible there is a body of literature on regulatees' perspectives on regulation that were not included as there was no attempt to link these perspectives to compliance.

## Quality appraisal

**Quantitative studies.** There were mixed findings in terms of the quality of quantitative studies (S1 File). Most followed best practice in terms of defining their samples, identifying and controlling for confounding factors, and using appropriate statistical methods. However, there were limitations evident in the use of self-reported measures (e.g. measurements of culture) and therefore it was not possible to be assured that these were objective.

**Qualitative and mixed methods studies.** Most qualitative and mixed methods studies were of good quality (S1 File). There was generally a congruence between the methods employed by the researchers and their objectives, data collection methods, analysis and interpretation of results. Common weaknesses in these studies tended to be a lack of reflexivity or the absence of a statement locating the authors culturally and theoretically in the research.

## Mapping findings to the CFIR

We coded findings to all five CFIR domains. We coded findings to 19 of the 39 CFIR constructs, in addition to coding findings to an additional construct for 'other' in each of the five domains.

## Intervention characteristics

This domain covers "aspects of an intervention that may impact implementation success, including its perceived internal or external origin, evidence quality and strength, relative advantage, adaptability, trialability, complexity, design quality and presentation, and cost" [30]. We coded nothing to four of the eight constructs in this domain: intervention source; trialability; design quality and packaging; cost. In total, 16 studies featured in this domain, see Table 2.

**Evidence strength & quality.** *Stakeholders' perceptions of the quality and validity of evidence supporting the belief that the innovation will have desired outcomes.*

Hospital staff expressed scepticism that a regulatory rating reflected their own perception of quality: "The sentiment seemed to be shared among others in the room, who. . .agreed that there was a tangible disconnect between their delivery of clinical care and the ratings and scores representing their care to the public" [18].

**Relative advantage.** *Stakeholders' perception of the advantage of implementing the innovation versus an alternative solution.*

In this instance, regulatory standards were the innovation to be implemented. Findings were coded here if they were concerned with advantages (or disadvantages) derived from complying. NH inspectors noted that enforcing closure of unauthorised NH would mean loss of bed capacity, thereby questioning the logic of implementing that solution [32]. The inspectors reasoned that turning a blind eye was somehow more desirable than the closure of thousands of

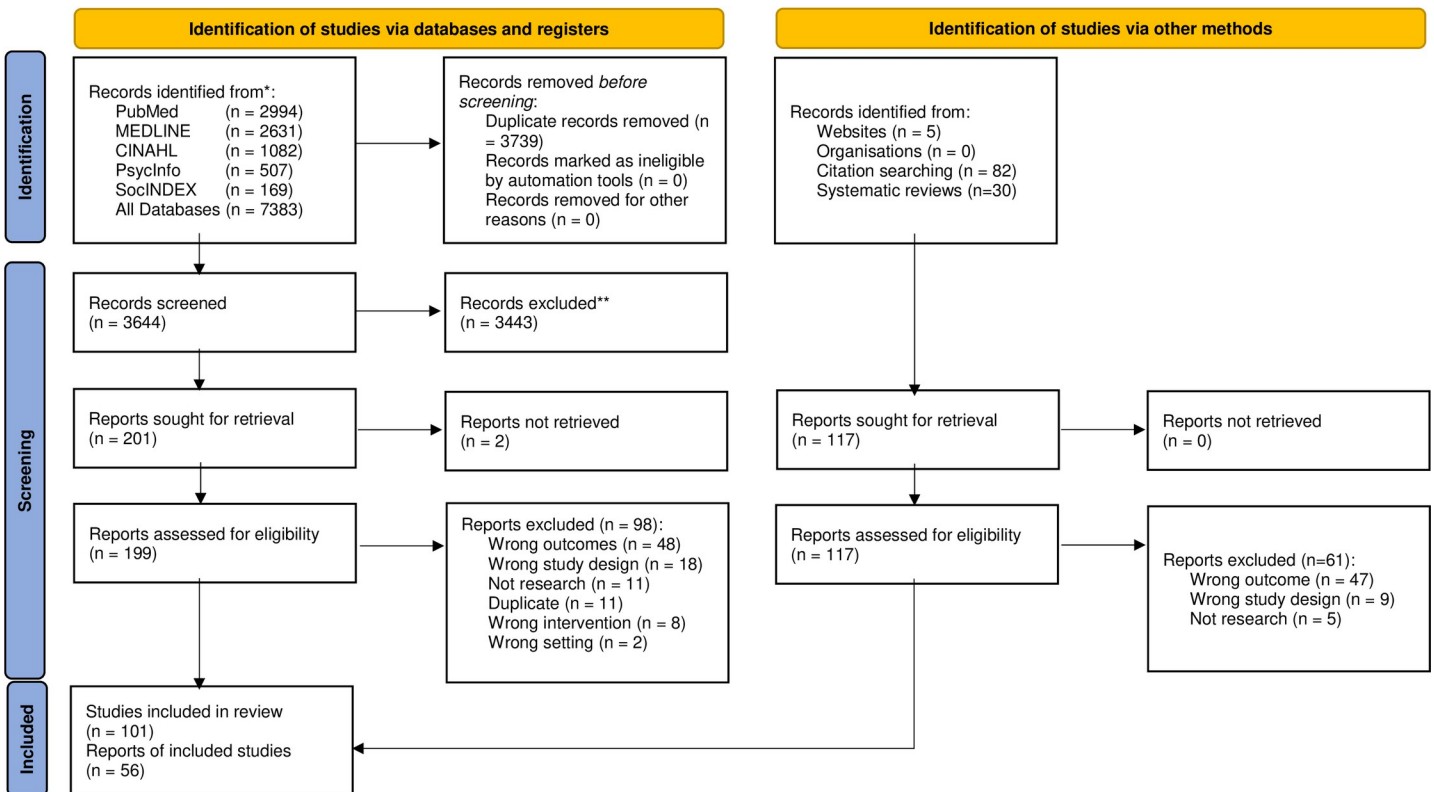

**Fig 1. PRISMA 2020 flow diagram: Determinants of regulatory compliance in health and social care services: A systematic review using the consolidated framework for implementation research.**

beds [32]. In a participant observation study, regulators provided "a veil" of compliance to care providers who then leveraged this to attract service users [31]. Managers "reproved workers not on the basis of whether tasks had been completed, but according to whether files reflected regulatory standards" [31]. Thus, management perceived the advantage of good documentation—even going as far as to falsify records—in order to achieve compliance [31].

**Adaptability.** *The degree to which an innovation can be adapted, tailored, refined, or reinvented to meet local needs.*

This construct refers to how regulatory measures can be adapted to fit with existing work practices. In a qualitative study, the requirement for clinicians to document secondary tasks to ensure compliance was reported to be difficult to adapt to a hospital's workflow [18]. Thus, it was challenging to comply because it created additional work or necessitated a work-around.

**Complexity.** *Perceived difficulty of the innovation, reflected by duration, scope, radicalness, disruptiveness, centrality, and intricacy and number of steps required to implement.*

A regulatory measure related to sepsis treatment procedures presented "unprecedented complexity"; study participants suggested it was so complex it required its own quality assurance meeting [18].

**Other.** *Findings were placed in this section where they could not be coded to an appropriate construct but were considered relevant to this domain.*

**Table 1. Characteristics of studies included in the systematic review.**

| Publication type | n (%) |
|---|---|
| Journal Article | 137 (87.3) |
| Thesis | 19 (12.1) |
| Book Section | 1 (0.6) |
| **Study setting** | **n (%)** |
| Nursing Homes | 143 (91.1) |
| Hospitals | 7 (4.4) |
| Nursing/Care Homes | 2 (1.3) |
| Assisted-living facilities | 2 (1.3) |
| Nursing Home/Assisted-Living Facilities | 2 (1.3) |
| Pharmacies | 1 (0.6) |
| **Study design** | **n (%)** |
| *Quantitative* | |
| Cross-sectional/Observational | 140 (89.2) |
| Longitudinal | 3 (1.9) |
| RCT | 2 (1.3) |
| Experimental | 2 (1.3) |
| *Qualitative* | |
| Case study | 4 (2.5) |
| Ethnography | 1 (0.6) |
| Focus groups | 1 (0.6) |
| *Mixed methods* | 4 (2.5) |
| **Country** | **n (%)** |
| USA | 140 (89.2) |
| Australia | 9 (5.7) |
| Europe | |
| United Kingdom | 3 (1.9) |
| Denmark | 1 (0.6) |
| Portugal | 1 (0.6) |
| Netherlands | 1 (0.6) |
| China | 1 (0.6) |
| Saudi Arabia | 1 (0.6) |

A study on attitudes to regulation reported that physicians who perceived a persuasive approach by regulators tended to be more compliant, as compared with a punishment approach [43]. Disapproving and re-integrating (i.e. respectful disapproval and then terminating disapproval through forgiveness upon improvement) inspection teams and compliance were positively associated whereas tolerant and stigmatizing teams were negatively associated [33]. Inspection teams that viewed intervention as necessary were positively associated with compliance, compared to three other types (cooperative; deterrence; persuasion & education) where there was a null association [41].

Inspection team characteristics (e.g. office base, team composition) and compliance was reported to be associated in all five studies that investigated these determinants [34–38]. The number of inspectors on a team was reported to have a null association with compliance [45]. Two studies investigated various tools used for inspection and both had an association with compliance [39, 40]. The type of inspection (e.g. inspections that mandated interviews with

**Table 2. Determinants coded to the intervention characteristics domain.**

| Construct | Determinant |
|---|---|
| **Intervention Source** | No studies |
| **Evidence Strength & Quality** | "The sentiment seemed to be shared among others in the room, who laughed and nodded, and agreed that there was a tangible disconnect between their delivery of clinical care and the ratings and scores representing their care to the public" <br> "As one nurse manager put it in a meeting," . . . it's not about the clinical management of sepsis, it's about fully reporting the "cookbook" steps of the goal" <br> "clinicians repeatedly defined metric compliance as distinct to their clinical practice or scientific expertise" [18] |
| **Relative Advantage** | "Rather than containing the worst aspects of corporate activity in this sector, regulation may serve to obscure what might be widespread routine forms of injurious activity through a veil of bureaucratic compliance" [31] |
| | "During the discussion one of the participants (E8) mentioned that there are an estimated 3,000 non-licensed nursing homes in Portugal. "An inspection with a global closing order of all the illegal nursing homes would create an unsolvable problem: what would become of all the elderly who live there nowadays, who are roughly 25,000 to 30,000 people? There are simply not enough beds for so many people" (E8). Therefore, as mentioned by E5: "Unfortunately there is a lot of covering-up and impunity in Portugal" [32] |
| **Adaptability** | "A mandate that requires clinicians to timestamp and document secondary tasks, just to ensure metric compliance, historically is not typical for ED workflow" [18] |
| **Trialability** | No studies |
| **Complexity** | "An internist involved in quality assurance, explained that clinicians and administrators agreed SEP-1 was so complex it would need its own meeting" <br> "His explanation echoes accounts from research at other hospitals, which confirm unprecedented complexity contributing to SEP-1's difficulty" [18] |
| **Design Quality & Packaging** | No studies |
| **Cost** | No studies |
| **Other** | Disapproving inspection teams [33] <br> Inspection team characteristics [34–38] |
| | Inspection survey tool [39, 40] <br> Inspection team: cooperative [41] <br> Inspection team: deterrence [41] <br> Inspection team: persuasion and education [41] <br> Inspection team viewed intervention as necessary [41] <br> Inspection type [42] <br> Perceived persuasive approach by regulator [43] <br> Perceived punishment approach by regulator [43] <br> Re-integrating inspection teams [33] <br> Stigmatizing inspection teams [33] <br> Tolerant inspection teams [33] <br> Unannounced inspections [44] |

service users) was reported to be associated with compliance [42] whereas unannounced inspections were reported to have a null association when compared with announced [44].

## Outer setting

The outer setting domain is defined as "external influences on intervention implementation including patient needs and resources, cosmopolitanism or the level at which the implementing organization is networked with other organizations, peer pressure, and external policies and incentives" [30]. Seventy-two studies were featured in the 'peer pressure' and 'external policy and incentives' constructs, as well as in 'other'. No determinants were coded to two of four constructs in this domain: 'cosmopolitanism' or 'needs and resources of those served by the organisation', see Table 3.

**Table 3. Determinants coded to the outer setting domain.**

| Construct | Determinant |
|---|---|
| Needs & Resources of Those Served by the Organization | No studies |
| Cosmopolitanism | No studies |
| Peer pressure | Higher competition [13, 16, 46–63]<br>Higher excess bed capacity in county [16, 64]<br>Higher service demand [62]<br>Nursing homes per population [64]<br>Proportion of chains in region [55, 65, 66]<br>Proportion of for-profit facilities [66]<br>Proportion of hospital-based facilities [55, 66]<br>Transparency (publication of star-rating) [53] |
| External Policy & Incentives | Higher state focus on infection prevention [67]<br>Minimum staffing standard [68–70]<br>More stringent nursing home administrator licensing criteria<br>Ombudsman available [71–73]<br>State-level mandatory overtime laws [46]<br>States with certificate of need law [74] |
| Other | Ageing advocacy group influence [66]<br>Democratic party controlled legislature (USA) [66]<br>Governor's institutional power [66]<br>Higher local income [47, 50, 52, 59, 60, 62, 63, 66]<br>Higher local unemployment [75]<br>Higher mean age of service users [51, 66, 76]<br>Higher population density [36, 46, 51, 62, 76–87]<br>Higher proportion of African-American service users [13, 52, 76, 88, 89]<br>Higher proportion of African-Americans residents in county [52]<br>Higher proportion of Asian service users [52]<br>Higher proportion of female service users [45, 51]<br>Higher proportion of George W. Bush supporters in county [74]<br>Higher proportion of Hispanic service users [13, 52, 89]<br>Higher proportion of Hispanic residents in county [52]<br>Higher proportion of married service users [45]<br>Higher proportion of other non-White service users [13]<br>Higher proportion of service users with dementia [14, 47, 49, 50, 54, 58, 59, 62, 63, 90–92]<br>Higher proportion of service users with a disability [16, 45, 82]<br>Higher proportion of service users with intellectual disability [49, 50, 54, 58, 59, 63, 92]<br>Higher proportion of service users with incontinence needs [14, 54, 82, 90, 91, 93–97]<br>Higher proportion of service users with mental illness [14, 47, 49, 50, 54, 57–59, 62, 63, 82, 90, 92–96, 98–102]<br>Higher proportion of service users with obstructive bowel syndrome [82]<br>Higher proportion of service users with high/complex care needs [14, 16, 46, 47, 49–52, 54, 58–64, 72, 82, 86, 90–97, 99, 103–110]<br>Higher proportion of veterans [111]<br>Higher proportion of White residents [51, 76, 112]<br>Higher public funding [47–50, 55, 57–60, 63–66]<br>Higher rate of religiosity [74]<br>Individualistic state [74]<br>Higher legislative professionalism [66]<br>Nursing home industry influence [66]<br>Moralistic state [66]<br>State with Democratic Governor (USA) [55, 66] |

**Peer pressure.** *Mimetic or competitive pressure to implement an innovation, typically because most or other key peer or competing organizations have already implemented or are in a bid for a competitive edge.*

Market pressure was frequently studied and we mapped related determinants to this construct. This may occur in two ways: pressure to have better compliance rates than competitors in order to attract more clients; or pressure to match or exceed the compliance levels of peer organisations (e.g. a chain of hospitals owned by one corporation). Of 20 studies investigating competition, four reported high competition was positively associated with compliance [16, 57–59], four had a negative association [60–63], 11 reported a null association [13, 46–55]; one had a u-shaped finding where higher competition and monopoly were both positively associated whereas moderate competition was negatively associated [56].

Other studies investigated NH markets in a region. A higher proportion of chain NHs in a state and compliance was negatively associated in two studies [55, 65], a third reported a null association [66]. The proportion of for-profit [66] and/or hospital-based NHs [55, 66] and compliance had a null association. The number of NHs in a county per population and compliance had a null association [64]. Additional findings related to this construct included: publication of quality ratings, which we construed as increasing competitive pressure within local markets (null [53]), excess bed capacity in a county (positive [64]; null [16]) and higher service demand (positive [62]).

**External policy and incentives.**   *A broad construct that includes external strategies to spread innovations including policy and regulations (governmental or other central entity), external mandates, recommendations and guidelines, pay-for-performance, collaboratives, and public or benchmark reporting.*

We interpreted this construct as relating to the regulatory environment, either via funding, legislative measures or quality improvement initiatives from external sources. In American states that had legislation/policies governing quality issues (e.g. minimum staffing standards [68–70] or infection control initiatives [67]) a 'spillover' effect was observed: service providers were more compliant with other regulations compared with states that had no such legislation/policies. Other findings included: presence of an ombudsman (positive [71]; null [72, 73]); stricter licensing criteria for NH managers (null [66]); mandatory overtime laws (negative [46]); and states with 'certificate of need' laws (positive [74]).

**Other.**   *Findings were placed in this section where they could not be coded to an appropriate construct but were still considered relevant to this domain.*

Most of the material listed below relates to service user needs or the characteristics of a given population. We coded measures of service user case-mix to the 'outer setting' domain because, although case-mix is treated within the 'inner setting', it is a measure of the broader population being served.

In the context of people using services, 43 studies investigated the proportion of service users with needs that may require more care (e.g. dementia, incontinence, mental ill health) [14, 16, 45–47, 49–52, 54, 57–64, 72, 82, 86, 90–110, 113]. In general, higher proportions of such service users and compliance was negatively associated. For example, a higher proportion of service users with higher or more complex care needs (typically measured by case mix or activities of daily living scores) and compliance was reported to be positively associated in five studies [14, 58, 90, 93, 97], negatively in 18 studies [14, 16, 46, 47, 49, 60, 62, 82, 90–93, 96, 99, 104, 105, 108, 113], with 21 reporting a null association [50–52, 54, 59, 61, 63, 64, 72, 82, 86, 92–94, 96, 97, 103, 106, 107, 109, 110] (some studies investigated multiple different measures of care needs and reported different associations with compliance. Therefore, these studies appear across more than one of the above categories).

Higher mean age of service users and compliance was negatively associated in two [66, 76] of three studies. Studies also investigated service user ethnicity: higher proportion of white

**Table 4. Determinants coded to the inner setting domain.**

| Construct | Determinant |
|---|---|
| Structural Characteristics | Continuing care retirement community [71, 92, 93]<br>County medical care facility [106]<br>Facility certification status [36]<br>Facility certification type [78, 87]<br>Facility provides skilled/specialist services [36, 62, 71, 80, 82, 91, 92, 95, 110, 115–117]<br>For-profit facility [13, 16, 32, 35, 36, 45, 47–52, 54, 57–65, 71, 72, 74, 76, 80–87, 90–93, 95, 103, 104, 106, 108, 113, 115–123]<br>Geographic location [13, 40, 45, 54, 61, 64, 79, 85, 87, 92, 94, 106, 117, 124, 125]<br>Government-owned nursing home without a shared owner in hospital referral region [93]<br>Higher number of years of facility in operation [45, 53, 93, 110, 115]<br>Hospital-based facility [14, 16, 36, 61, 62, 71, 74, 78, 91, 93, 95, 104, 106, 126]<br>Hospital participating in a clinical surgery registry [105]<br>Larger size facility [13, 14, 16, 34–37, 45–47, 49–52, 54, 57–64, 71, 72, 74, 77, 78, 80–83, 85–87, 91–97, 105, 106, 108, 113, 114, 116–118, 120–124, 126–128]<br>Level-1 trauma centre [105]<br>Major teaching hospital [105]<br>Model of ownership [62, 77, 78, 92, 96, 97, 107, 123, 126, 129, 130]<br>Number of years organisation has engaged with accreditation [121]<br>Nursing home co-location [127]<br>Nursing home located near psychiatric hospital [102]<br>Nursing home participating in community nursing home programme [111]<br>Nursing home with shared owner of another nursing home in hospital referral region [93]<br>Nursing home with specialist registration [110]<br>Revenue source [78]<br>Safety-net hospital status [105]<br>"social workers and admission coordinators [in poorly-performing nursing homes as measured by compliance] were instructed to fill beds with Medicare residents before accepting Medicaid residents" [131] |
| Networks & Communications | Chain membership [13, 47, 49–52, 54, 58–63, 74, 76, 80, 82, 90, 92–96, 104, 106, 108, 113, 116, 122, 123, 132]<br>Higher facility consultation with families [52, 76, 90, 116]<br>Higher facility consultation with residents [71, 90, 116]<br>Higher staff meeting frequency [133, 134]<br>Higher use of information technology [81, 124, 134–136]<br>Notification method for residents with potential infection [133]<br>Nursing home has resident/family council [93]<br>Nursing home is associated with other compliant facilities [137]<br>Nursing homes that use a satisfaction survey [138]<br>Process standardisation across chain [94]<br>"In the historically high-performing nursing homes (NH1 and NH2), administrative and direct care staff used multiple modes of communication to care for residents and their families" [131]<br>"In both homes, the flow of information, both hierarchically and laterally, was important to meeting resident needs quickly and thoroughly" [131]<br>"In the low-performing nursing homes (NH3 and NH4), the conflicting message regarding the organizational mission, that is, the explicit (resident care) versus the implicit (economic viability and regulatory compliance), fragmented and confused staff" [131]<br>"Problem resolution resulted in the creation of new forms, additional steps, problem intensification, and communication breakdown within and across disciplines and departments" [131] |
| Culture | Changing from non-profit to for-profit facility [139]<br>Higher culture change focus [140, 141]<br>Higher focus on reducing hospitalisation [133]<br>Higher focus on patient-centred care [103, 108, 142, 143]<br>Nursing home strategic focus [144]<br>Organisational disposition towards regulation [41, 137] |
| Implementation Climate | |

(*Continued*)

**Table 4.** (Continued)

| Construct | Determinant |
|---|---|
| Tension for Change | More strict enforcement [53, 93, 115, 145–148]<br>"participants expressed an awareness of the pressure they feel to reduce the overall reported rate to fulfill regulatory requirements and maintain or improve their star rating via the CMS public-reporting system to denote nursing home care quality" [149] |
| Compatibility | "The powerplan was stressed as the key to correct documentation and thereby improved compliance with the metric"<br>"a coordinated, and inevitably, standardized approach to treating sepsis based on both policy requirements and clinical expertise. They also produce the documentation necessary for meeting and reporting on regulatory sepsis metrics" [18] |
| Relative Priority | Provider concern about regulatory enforcement [45] |
| Organizational Incentives & Rewards | No studies |
| Goals & Feedback | No studies |
| Learning Climate | "Staff across and within all disciplines [in high-performing nursing homes as measured by compliance] were involved in problem solving and decision making; staff were reminded that decision making and creativity improved with multiple perspectives" [131] |
| Readiness for Implementation | No studies |
| Leadership Engagement | Accredited facility [105, 118, 150–152]<br>Higher ownership turnover [84]<br>Receipt of a quality award [153]<br>"In the high-performing homes, the leadership behaviors of the NHA and the DON created a clear, explicit, and coherent mission, a strong sense of purpose, for the organization. . . ..The importance of leadership relationship behaviors and the congruence of the stated and lived mission of the home were strikingly different between high- and low-performing homes" [131] |
| Available Resources | No studies |
| Access to Knowledge & Information | No studies |
| Other | **Staffing**<br>Higher administrative staffing [14, 54]<br>Higher agency/contract staff levels [46, 143, 154, 155]<br>Higher degree of nursing centralisation in a nursing home [36]<br>Higher food-service staffing [156]<br>Higher housekeeping staffing [14, 156]<br>Higher infection control professional quality [134]<br>Higher job satisfaction [157]<br>Higher level of certified medication aide staffing [158]<br>Higher level of staff training [91, 94, 110, 116, 133]<br>Higher level of trainee nurse aides [158]<br>Higher licensed practitioner/vocational nurse staffing [14, 47, 49–51, 54, 57–61, 63, 77, 81, 82, 93, 95, 108, 114, 123, 143, 156, 158]<br>Higher nurse aide absenteeism [132]<br>Higher nurse aide staffing [14, 15, 34, 36, 47, 49–51, 54, 57–61, 63, 81, 82, 93, 95, 108, 114, 123, 126, 127, 143, 156, 158, 159]<br>Higher ratings for work effectiveness and practice environments in nursing homes [108, 120]<br>Higher ratio of registered nurses as against other nurse and care staff [35, 57, 59, 97, 109, 113, 143, 160]<br>Higher registered nurse staffing [14, 15, 35, 49–51, 54, 58, 59, 61, 63, 66, 93, 95, 96, 107–109, 113, 114, 122, 123, 127, 143, 154, 156, 158, 159, 161]<br>Higher social services director caseload level [84]<br>Higher specialist staffing [14, 34, 36, 57, 91, 116, 156, 162]<br>Higher staff turnover [35, 52, 54, 76, 97, 133, 134, 143, 154, 160, 163–169] |

(*Continued*)

**Table 4.** (Continued)

| Construct | Determinant |
|---|---|
| | Higher staff workload [84, 133]<br>Higher total nursing staff [35, 36, 47, 57, 60, 61, 72, 81, 83, 105, 107, 126, 158, 161, 164]<br>Higher total staffing [52, 91, 94, 116, 122, 154]<br>Nursing home meeting minimum staffing standard [60, 161]<br>"Administrative leadership behaviors that fostered staff appreciation included routine practices such as having adequate staffing levels and resources to do the job (e.g., resident lifts); helping out on the floor by making beds, passing meal trays, and assisting residents when needed" [131]<br>**Financial**<br>Facilities that provided finances training, meetings or expert consultation related to infection control [133]<br>Higher facility costs [64, 76, 82, 97, 106, 130, 170–173]<br>Higher facility income [52, 64, 110, 172]<br>Higher facility market share [174]<br>Higher facility profitability [13, 51, 97, 175, 176]<br>Higher staff salary levels [128]<br>Higher proportion of private payers [62, 97, 116]<br>Higher proportions of publicly-funded service-users [13, 14, 16, 35, 36, 46–52, 54, 57–59, 61, 62, 64, 66, 76, 77, 80, 82, 86, 87, 90–92, 95, 97, 104, 106, 107, 109, 110, 113, 114, 116]<br>**Admissions**<br>Accepts Medicaid admissions [35, 71, 80, 83, 115]<br>Accepts Medicare admissions [71, 104]<br>Higher admission rate [82, 96]<br>Hospitals accepting critical access or rural referrals [86]<br>Higher occupancy levels [16, 46, 47, 49–51, 54, 55, 57–61, 63–65, 82, 90–93, 95, 106, 110, 113, 119, 126, 154]<br>**Ownership**<br>Recently-acquired nursing homes [93, 177]<br>Poor quality nursing homes that were acquired by a chain [177]<br>Higher quality of acquiring nursing home chain [177]<br>**Infection control**<br>Additional job responsibilities for infection control staff<br>Participation in infection control collaborative [133, 134]<br>Types of resources used to determine and treat infection [133]<br>**Chains**<br>Physical plant standardisation across a nursing home chain [94] |

service users (positive [76]; null [51]); higher proportion of Asian service users (positive [52]); higher proportion of black service users (negative [76, 88]; null [13, 52, 89]); higher proportion of Hispanic service users (null [52, 89]; positive [13]); higher proportion of other non-white service users (positive [13]). Studies also reported on proportion of female service users (positive [45]; null [51]), proportion of married service users (positive [45]), and proportion of service users that were war veterans (negative [111]).

Other studies measured characteristics of the broader population served by a provider. Higher population density and compliance was positively associated in five studies [62, 76, 84, 85, 87], negatively in one [86], with ten finding a null association [36, 46, 51, 77–83]. Other material featuring in this construct were the proportion of elderly in the population (positive [50, 60, 74], negative [49, 55, 58, 63], null [47, 59]); higher per-capita/household income (positive [49, 66]; negative [46, 58]; null [47, 50, 52, 59, 60, 62, 63]) higher unemployment rates (positive [75]); higher rates of religiosity (positive [74]); higher proportions of African-Americans in a county (positive [52]); and higher proportions of Hispanics in a county (positive [52]).

Higher levels of public/state funding for services and compliance was positively associated in six studies [47, 50, 57, 58, 60, 114], negatively in one study [63] and seven studies reported no association [48, 49, 55, 59, 64–66].

Three studies from the United States examined political context such as having a Democratic party Governor (negative [55]; null [66]); a higher level of legislative professionalism [66]; the proportion of people that supported George W. Bush (positive [74]); having a Democratic-controlled legislature (negative [66]); or states designated as individualistic (positive [74]) or moralistic (null [74]). The presence of lobbying groups and compliance was reported to have a null association [66].

## Inner setting

The majority of studies (n = 132) included at least one determinant coded to this domain (see Table 4). We coded no determinants to five out of 12 constructs: implementation climate–organisational incentives and rewards; implementation climate–goals and feedback; readiness for implementation; readiness for implementation–available resources; readiness for implementation–access to knowledge and information.

**Structural characteristics.** *The social architecture, age, maturity, and size of an organization.*

We construed this construct as including a wide range of organisational characteristics such as size, location, for-profit status and services provided.

Of the 15 studies that evaluated geographic location (comparing cities or regions), 14 reported compliance differed depending on location [13, 40, 45, 54, 61, 64, 79, 85, 87, 92, 94, 106, 117, 125]; one [124] reported a null association.

For-profit service providers and compliance were negatively associated in 29 studies [13, 16, 32, 45, 48, 52, 57, 60–62, 65, 72, 74, 76, 80, 84–87, 90–93, 104, 113, 115, 116, 119, 123], positively associated in seven [50, 58, 63, 81, 95, 117, 121], with a null association in 17 studies [35, 36, 47, 49, 51, 54, 59, 64, 71, 82, 83, 103, 106, 108, 118, 120, 122]. Aligned to the profit motive, in a qualitative study, staff in poorly-performing NHs (as measured by compliance) were instructed to prioritise the admission of people that attracted higher fees [131].

Larger facilities and compliance were negatively associated in 40 studies [13, 14, 16, 34–37, 46, 50–52, 54, 57, 58, 60–62, 64, 74, 77, 80, 82, 83, 86, 91–96, 106, 113, 114, 116, 117, 120, 122, 124, 127, 128]; three reported a positive association [59, 63, 71] and 16 a null association [45, 47, 49, 72, 78, 80, 81, 85, 87, 97, 105, 108, 118, 121, 123, 126] (one study [80] reported larger size NHs had a null association whereas larger size assisted-living facilities had a negative association, thereby represented in both columns). The longer a service was in operation was reported to be associated with poorer compliance in four of five studies [45, 53, 93, 110, 115]. Hospital-based NHs and compliance were positively associated in five studies [16, 62, 91, 93, 95], negatively in one [74], and there was no association in eight studies [14, 36, 61, 71, 78, 104, 106, 126].

There was no consistent association with compliance for facilities that provided a range of skilled/specialist services (positive [62, 71, 80, 82, 91, 116]; negative [62, 91, 92, 110, 115–117]; null [36, 62, 92, 95]). Assisted-living facilities with a nursing licence and compliance were negatively associated [80]; those with a mental health care licence positively associated [80]; and those providing memory care services had a null association [80]. Various models of facility ownership (e.g. publicly-owned, voluntary organisation or private corporation) and compliance were reported to be associated in five studies that investigated this determinant [78, 96, 123, 129, 130], whereas six reported a null association [62, 77, 92, 107, 126, 128].

Among other findings were: continuing care retirement communities (positive [92, 93]; null [71]); location beside a psychiatric hospital (negative [102]); county medical care facilities (null [106]); participation in a community NH program (negative [111]); teaching hospital status (negative [105]; null [86]); NH location (in the context of being co-located with various other types of services) was associated with compliance [127]; revenue source (null [78]); facility certification type was associated with compliance in one study [78] with a null association reported in another [87]; facilities in the United States designated as safety-net hospitals (i.e. serve all populations regardless of insurance status) (negative [105]); hospitals designated as level-1 trauma centres (negative [105]); hospitals participating in a clinical surgery registry (negative [105]); facility certification status (null [36]); facility proximity to a psychiatric hospital (negative [102]); nursing home shared ownership status (a form of public-private ownership) (positive [93]); and more organisational experience with the accreditation process (positive [121]).

**Networks and communications.** *The nature and quality of webs of social networks, and the nature and quality of formal and informal communications within an organization.*

Chain facilities and compliance had a negative association in 14 studies [13, 58–60, 62, 74, 80, 90, 95, 104, 113, 116, 122, 132], eight reported a positive association [47, 50, 63, 76, 93, 94, 96, 123] and nine a null association [49, 51, 52, 54, 61, 82, 92, 106, 108]. NHs that had a history of compliance were found to have effective communication practices, both hierarchically and laterally in a qualitative study: "administrative and direct care staff used multiple modes of communication to care for residents and their families" [131]. In contrast, poorly-performing nursing homes "confused staff" with conflicting messages regarding the service's organisational mission and tension between providing resident care and ensuring economic viability [131].

A range of other factors were found to have no clear association with compliance: higher consultation with families (positive [76, 116]; negative [76]; null [52, 90]); higher consultation with NH residents (positive [116]; negative [90]; null [71]); NHs with family/resident councils (negative [93]); NHs that used a satisfaction survey (positive [138]); higher use of information technology (positive [81, 136]; null [124, 134, 135]); higher staff meeting frequency (negative [134]; null [133]); administrative/clinical standardisation across a nursing home chain (null [94]); facility methods for communicating potential IC issues (null [133]); maintaining a list of service users with infection (null [133]); and association with other compliant facilities (null [137]).

**Culture.** *Norms, values, and basic assumptions of a given organisation.*

In the context of our research question we interpreted this construct to refer to institutional culture with an obvious relevance to regulatory compliance. A study of motivational postures towards regulation reported that facilities adopting a posture of disengagement and compliance were negatively associated whereas those with a culture of resistance to regulation were positively associated with compliance [41]. The authors surmised that this seemingly incongruous finding may be explained by an interaction with the inspectors' regulatory style [41]. Two further determinants in this study were reported as having a null association with compliance: a culture of capture (high degree of cooperation with regulators) and of managerial accommodation (where management were cooperative and accepted responsibility for implementation) [41]. A similar study measured the perspectives of senior managers in NHs in terms of whether they 'believed in the standards' or exhibited a 'subculture of resistance"; both were positively associated with compliance [137].

Organisations with a higher patient-centred culture and compliance were positively associated in three studies [103, 108, 143], with a fourth reporting a null association [142]. NHs that had a "culture change" focus and compliance were positively associated in two studies [140, 141]. One study investigated whether a NH's culture of cooperation with a regulator was associated with compliance (null [137]).

Other factors included a culture that focused on reducing hospitalisation for residents (negative [133]) and culture change brought on by converting from a non-profit to for-profit facility (negative [178]). The strategic focus of a NH (e.g. attracting clients through higher quality/lower costs) was associated with compliance [144].

**Implementation climate–tension for change.** *The degree to which stakeholders perceive the current situation as intolerable or needing change.*

We interpreted this construct as including studies that investigated organisational drivers for improved compliance or the urgency with which changes in compliance levels were considered necessary by organisations. A qualitative study on efforts to reduce anti-psychotic medication in NHs reported staff felt pressure to fulfil regulatory requirements and improve their publicly-reported quality level [149]. There were mixed findings on the stringency of regulators: three studies [53, 115, 147] reported stricter enforcement and compliance was positively associated, four negatively [93, 115, 145, 146], and one [148] reported a null association (one study [115] reported determinants in the positive and negative categories).

**Implementation climate–compatibility.** *The degree of tangible fit between meaning and values attached to the innovation by involved individuals, how those align with individuals' own norms, values, and perceived risks and needs, and how the innovation fits with existing workflows and systems.*

We took compatibility to mean the degree to which a regulatory measure fits with pre-existing practices in an organisation. One qualitative study featured here, where the author reported compliance was improved because the regulatory metric was compatible with the electronic health record system enabling "a coordinated. . .standardized approach to treating sepsis based on both policy requirements and clinical expertise. They also produce the documentation necessary for meeting and reporting on regulatory sepsis metrics" [18].

**Implementation climate–relative priority.** *Individuals' shared perception of the importance of the implementation within the organization.*

This construct relates the implementation of regulations to how important a goal it is for those charged with achieving compliance. One study investigated a range of issues relating to an organisation's perception of regulation, their estimation of the likelihood of non-compliance being detected, and the perceived severity of regulatory sanctions [45]. The perceived severity of withholding a funding increase and compliance was negatively associated; the remaining factors in this study (e.g. probability of: prosecution/state detection/withdrawal of licence) had a null association [45].

**Implementation climate–learning climate.** *A climate in which: 1. Leaders express their own fallibility and need for team members' assistance and input; 2. Team members feel that they are essential, valued, and knowledgeable partners in the change process; 3. Individuals feel psychologically safe to try new methods; and 4. There is sufficient time and space for reflective thinking and evaluation.*

The authors of one qualitative study on high- and low-performing nursing homes (as measured by compliance levels) reported that high-performing nursing homes were characterised by management who actively encouraged staff participation in decision-making [131].

**Readiness for implementation–leadership engagement.** *Commitment, involvement, and accountability of leaders and managers with the implementation of the innovation.*

Organisations where management submit to an external quality assessment (e.g. voluntary accreditation) were more likely to be compliant in all four studies that investigated this [150–153]. A fifth study investigated hospital accreditation by the Joint Commission (negative [105]) and the Commission on Cancer (null [105]). The authors of a qualitative study reported a difference in leadership behaviours when comparing high- and low-performing NHs: "the leadership behaviours of the NHA [Nursing Home Administrator] and the DON [Director of Nursing] created a clear, explicit, and coherent mission, a strong sense of purpose, for the organisation. The importance of leadership relationship behaviours and the congruence of the stated and lived mission of the home were strikingly different between high- and low-performing homes" [131].

**Other.** *Findings were placed in this section where they could not be coded to an appropriate construct but were still considered relevant to this domain. For example, while many studies may have investigated whether staffing levels were associated with compliance, they did not investigate staffing that were specifically dedicated to implementing regulations or achieving compliance.*

*Other*: *Staffing.* Of the 29 studies examining registered nurse staffing, 19 reported higher levels of this staff grade and compliance were positively associated [14, 35, 49–51, 58, 59, 61, 93, 95, 108, 113, 114, 123, 143, 156, 158, 159, 161], nine reported a null association [15, 54, 63, 66, 96, 109, 122, 127, 154] and one reported a negative association [107]. Likewise, total nurse staffing levels (all grades of nurse staff) and compliance were positively associated in eleven studies [35, 47, 57, 60, 81, 83, 107, 126, 158, 161, 164], with two finding a negative association [61, 83] and three a null association [36, 72, 105]. The results were similar for nurse aides/care assistants: 15 studies reported higher levels these staff and compliance were positively associated [14, 34, 36, 47, 49, 51, 59–61, 93, 95, 126, 127, 143, 156, 158], nine reported a null association [15, 54, 63, 81, 82, 108, 114, 123, 159] and three reported a negative association [50, 57, 58]. A higher ratio of registered nurses as against other nurse and care staff and compliance was positively associated in four studies [97, 109, 113, 143] with a null association in a further four [35, 57, 59, 160]. Higher levels of licensed practitioner/vocational nurses and compliance were positively associated in five [47, 49, 50, 67, 156], negatively in five [57–60, 77], and a null association reported in 13 [14, 51, 54, 61, 63, 81, 82, 93, 108, 114, 123, 143, 158].

Higher specialist staff levels (e.g. dietary professionals) and compliance were positively associated in five studies [14, 34, 91, 156, 162], negatively in one [116]; and null in two [36, 57] NHs meeting state criteria for minimum staffing and compliance were positively associated in two studies [60, 161].

Among other staffing-related determinants were the following: higher total staffing levels (positive [91, 116]; null [52, 94, 122, 154]); higher proportion of trained staff (positive [91, 133]; negative [116]; null [94, 110]); higher agency/contract staff levels (negative [46, 143, 155]; null [154]); higher administrative staff levels (positive [14]; null [54]); higher housekeeping staff levels (null [14, 156]); higher food-service staff levels (null [156]); higher infection control professional quality (measured by certification level and training attainment)(null [134]); higher levels of trainee nurse aides (null [158]); and higher levels of certified medication aide staffing (null [158]).

In a qualitative study, the authors found high-performing nursing homes (as measured by compliance levels) had leaders who made adequate resources available to allow staff do their job [131]. Higher staff workload was associated with poorer compliance in two studies [84, 133].

Of the 19 studies that assessed staff turnover, 16 reported high staff turnover and compliance was negatively associated [52, 54, 76, 119, 133, 143, 154, 160, 163–169, 179] and three reported no association [97, 133, 134] (one study [133] had two measures of turnover with different associations and is counted in both the negative and null columns). Higher job satisfaction and compliance had a null association in one study [157]; nurse-aide absenteeism and compliance was negatively associated [132]. Two studies found higher ratings for work effectiveness and practice environments in NHs and compliance were positively associated [108, 120]. The degree of nursing centralisation in a NH and compliance had a null association [36]. A higher social services director caseload level and compliance was negatively associated [84].

*Other*: *Financial*. The majority of studies listed in this section were literature from the United States. Higher costs (most of which were expressed as costs per patient day in a NH context) and compliance were negatively associated in six studies [64, 76, 82, 130, 171, 173], four reported a null association [97, 106, 171, 172], and one a positive association [170] (One study [171] had cost determinants in both the negative and null categories). Services with higher proportions of publicly-funded service-users and compliance also tended to be negatively associated (six positive [13, 47, 62, 76, 90, 109]; 21 negative [14, 16, 34, 48, 49, 52, 57, 58, 64, 77, 80, 90–92, 95, 97, 104, 110, 113, 114, 116]; 16 null [13, 35, 36, 46, 50–52, 54, 59, 61, 66, 82, 86, 87, 106, 107]) (some studies investigated multiple different types of public funding and reported different associations with compliance. Therefore, these studies appear across more than one of the above categories). Three studies investigated the proportion of private-payers (positive [62]; negative [116]; null [128]). Staff salary levels and compliance had a null association [97].

Four studies investigated facility profit level/margin (positive [51, 175]; negative [13]; null [97]). One study reported a u-shaped finding where average financial performance was positively associated with compliance whereas both high and poor financial performance were negatively associated [176]. Higher facility income and compliance was positively associated in two studies [52, 110], with two reporting a null association [64, 172]. Facilities that provided finances to attend infection control training and compliance were positively associated [133], whereas there was a null association when providing funding for infection control meetings or access to expert consultation on infection control [133]. Higher facility market share and compliance was reported to be positively associated [174].

*Other*: *Admissions*. NHs or assisted-living facilities accepting Medicaid (a public health insurance program in the United States for people on low incomes) admissions and compliance were positively associated in three studies [35, 71, 83], a fourth reported a negative association [115], and a fifth a null association [80]. NHs accepting Medicare (a health insurance program in the United States generally for those over 65 or with specific conditions) admissions and compliance were positively associated in two studies [71, 104]. NH admission rate and compliance had a null association in two studies [82, 96]. Hospitals accepting critical access or rural referrals and compliance had a null association [86]. Higher occupancy levels and compliance had no consistent direction of association in 28 studies (eight positive [16, 58, 59, 63, 93, 106, 119, 154]; nine negative [46, 55, 57, 60, 61, 65, 91, 95, 113]; 11 null [47, 49–51, 54, 64, 82, 90, 92, 110, 126]).

*Other*: *Ownership*. Recently-acquired NHs and compliance were reported to have a negative [177] and null [93] association. Compliance was reported to improve in poor quality NHs that

were acquired by a chain [177]. Where a NH was acquired by a chain, higher quality chains were positively associated with compliance [177].

*Other*: *Infection control.* Participation in an infection control collaborative focused on reducing MRSA and compliance was positively associated [133]; while other infection control collaborations had a null association [133, 134]. The types of resources used to determine and treat infection and compliance had a null association [133]. A study of additional job responsibilities for infection control staff reported that being Director/Assistant Director of Nursing and compliance was negatively associated, all other roles were reported as having a null association [133].

*Other*: *Chains.* Physical plant standardisation across a nursing home chain and compliance was positively associated [94].

## Characteristics of individuals

This domain reflects how "individuals' beliefs, knowledge, self-efficacy, and personal attributes" may impact the implementation of regulations. We found no determinants that could be coded to two of the five constructs in this domain: knowledge and beliefs about the innovation, and individual stage of change. In total, seven studies–the lowest for any domain–investigated a determinant coded to this domain (see Table 5).

**Self-efficacy.** *Individual belief in their own capabilities to execute courses of action to achieve implementation goals.*

Studies were included in this construct if they examined perceptions of capacity to implement regulations and its relationship to successful compliance. When nursing home managers felt that legitimate means of achieving compliance were blocked by owners, this was negatively associated with compliance [137].

**Individual identification with organisation.** *A broad construct related to how individuals perceive the organization, and their relationship and degree of commitment with that organization.*

Higher job commitment and compliance was positively associated in two studies [132, 157] and had a null association in a third [137].

Table 5. Determinants coded to the characteristics of individuals domain.

| Construct | Determinant |
|---|---|
| Knowledge & Beliefs about the Innovation | No studies |
| Self-efficacy | Blocked legitimate opportunities to comply [137] |
| Individual Stage of Change | No studies |
| Individual Identification with Organization | Higher job commitment [132, 137, 157] |
| Other Personal Attributes | Directors of nursing having a higher emotional connection with inspectors [137]<br>Higher level of task expectation (a measure of staff familiarity and clarity with their roles) [103] |
| Other | Higher staff empowerment [84]<br>Ranking of infection control activities in order of how time-consuming each were [133]<br>Senior managers' perception of what is important in job knowledge [180] |

**Table 6. Determinants coded to the process domain.**

| Construct | Determinant |
|---|---|
| Planning | No studies |
| Engaging | |
| Opinion Leaders | No studies |
| Formally Appointed Internal Implementation Leaders | No studies |
| Champions | No studies |
| External Change Agents | No studies |
| Key Stakeholders | Physician represented on the infection control committees [133]<br>"Administrative personnel made decisions without involving the staff or residents affected by the change…Administration handed down a solution, without input…the social work department director was distressed when she was not included in the problem solving and resolution of a survey deficiency" [131] |
| Innovation Participants | No studies |
| Executing | "The data analysis for this study revealed challenges from different non-clinical departments regarding pushing person centered care policies but showed little to no negative impact on annual state survey results. Most participants responded that the department had great annual survey results after achieving Eden Alternative certification" [181]<br>"One physician explained that she orders blood cultures on every patient of hers that comes through the emergency room because the sepsis metric requires that immediate cultures are documented" [18]<br>"The dynamics described in this article reveal that the formal aspects of caring—those enacted by regulatory authorities—are superseded by material conditions, which can lead standards to be enacted discursively but practically inactivated…What is uncovered in this study is that compliance with various standards becomes feigned by the corporation just as policing is performed by the so-called independent regulators…In the case of social care, it may be that the primary intention of regulation is to give credence to the overall system and the accumulation of private enterprises that takes place by establishing the appearance of regulatory effort coupled with an illusion of corporate compliance" [31]<br>"Participants recognized that there were many potential benefits of antipsychotic medication reduction with four primary themes: (d) Improvement in the facility Quality Indicator score (regulatory compliance)" [149] |
| Reflecting & Evaluating | "Clinicians felt inundated with pressure and performance evaluations that tracked their compliance with the sepsis metric" [18] |
| Other | Availability of corporate training across a nursing home chain [94]<br>Facility has a mechanism for providing hand hygiene feedback to staff [133]<br>Formal staff teams [108]<br>Hand hygiene product used in facility [133]<br>Higher degree of managerial control [45, 137]<br>Higher level of consensus leadership by Directors of Nursing [132]<br>Higher levels of antibiotic prescribing in nursing homes [133]<br>Higher levels of management autonomy [45]<br>Higher quality senior managers [76, 103, 110, 118, 133]<br>Higher scores on hand hygiene audits [182]<br>Methods to determine service user infection [133]<br>More time spent on infection control activities [133]<br>Providing quality indicator outcome information to nursing home managers [183]<br>Self-managed staff teams [108]<br>Training methods used for infection control [133] |

**Other personal attributes.**   *A broad construct to include other personal traits such as tolerance of ambiguity, intellectual ability, motivation, values, competence, capacity, and learning style.*

Any studies which reported on miscellaneous personal attributes and their association with compliance were coded here. Directors of nursing having a higher emotional connection with inspectors and compliance was positively associated [137]. A higher level of task expectation (a measure of staff familiarity and clarity with their roles) and compliance was negatively associated [103].

**Other.**   *Findings were placed in this section where they could not be coded to an appropriate construct but were still considered relevant to this domain.*

The degree to which staff felt empowered to do their job and compliance was positively associated [84]. A study of infection control attitudes among NH staff asked respondents to rank a range of activities in order of how time-consuming each were. The ranking of vaccination (negative [133]) and infection control policy development (positive [133]) as highly-time consuming and compliance was associated, the remainder having a null association. Senior managers' perception of what is important in job knowledge (e.g. leadership; resident care; human resources) and compliance had a null association [180].

## Process

The process domain refers to "stages of implementation such as planning, executing, reflecting and evaluating, and the presence of key intervention stakeholders and influencers including opinion leaders, stakeholder engagement, and project champions" [30]. No studies were coded to six of the 10 constructs within the domain: planning; engaging–opinion leaders; engaging–formally appointed internal implementation leaders; engaging–champions; engaging–external change agents; engaging–innovation participants. In total, 17 studies investigated a determinant coded here (see Table 6).

**Engaging–key stakeholders.**   *Individuals from within the organization that are directly impacted by the innovation, e.g., staff responsible for making referrals to a new program or using a new work process.*

In this construct, compliance can be influenced by how key individuals in an organisation are affected, or consulted with, during the ongoing process of implementing regulations. In a qualitative study, the compliance of NHs was poor where management were reported to make decisions without involving other staff and residents: "Administration handed down a solution, without input. . .the social work department director was distressed when she was not included in the problem solving and resolution of a survey deficiency" [131]. Including physicians on infection control committees and compliance was positively associated [133].

**Executing.**   *Carrying out or accomplishing the implementation according to plan.*

We regarded this construct quite broadly and included determinants that described how an organisation carried out its functions and which could reasonably be argued to have some bearing on the implementation of regulations.

Four qualitative studies feature in this construct. One author argued that the execution of regulation is a façade that masks poor quality care; a form of performative compliance in which both the regulator and regulatee are complicit in providing false public assurance [31]. A second qualitative study described how a physician developed a work-around for a metric: "ordering blood tests ahead of a confirmed sepsis diagnosis to ensure she doesn't fail the metric

later if the patient becomes septic" [18]. Compliance was also achieved as a consequence of implementing another intervention such as a reduction in anti-psychotic medication or the implementation of a more person-centred care model in NHs [149, 181].

**Reflecting and evaluating.** *Quantitative and qualitative feedback about the progress and quality of implementation accompanied with regular personal and team debriefing about progress and experience.*

Studies that investigated how the provision of feedback or information on the implementation of regulations impacted the process of implementation were coded here. The author of a qualitative study reported clinicians felt overburdened with performance evaluations, ultimately developing work-arounds and "gaming the system" in order to worry less about meeting the metric [18].

**Other.** *Findings were placed in this section where they could not be coded to an appropriate construct but were still considered relevant to this domain.*

A higher level of consensus leadership by Directors of Nursing and compliance was positively associated [132]. The availability of corporate training across a NH chain and compliance had a null association [94]. A higher degree of managerial control and compliance had positive [137] and null [45] associations in two different quantitative studies whereas higher levels of management autonomy and compliance were positively associated [45]. Having higher quality senior managers (as measured by educational attainment or years of experience) and compliance was positively associated in three studies [76, 103, 110], with two finding a null association [118, 133].

A study of infection control practices reported that, of five methods to determine resident infection, only one (using Centers for Disease Control definitions [184]) was positively associated with compliance, the remainder having a null association [133]. In the same study, whether a NH had a mechanism for providing hand hygiene feedback to staff, as well as the type of hand hygiene product used, and compliance, had null associations [133]. Training methods used for infection control, video-based training, face-to-face training and handouts/flyers and compliance all had null associations [133]. The time spent on infection control activities and compliance had a null association [133].

Self-managed staff teams and compliance were positively associated [108] whereas more formal staff teams had a null association [108]. Higher levels of antibiotic prescribing in nursing homes and compliance were negatively associated [133]. Higher scores on hand hygiene audits and compliance had a null association [182]. The provision of quality indicator outcome information to NH managers and compliance had a null association [183].

## Discussion

The evidence base indicates that the following structural characteristics were positively associated with regulatory compliance: smaller facilities (as measured by bed capacity); higher nurse staffing levels; lower staff turnover. A facility's geographic location was also strongly associated with compliance (Table 7). There was a paucity of evidence on the processes associated with successful implementation of regulations.

The evidence presented in this review was predominantly quantitative (n = 147, 94%), conducted in the United States (n = 140, 90%) and focused on NHs (n = 143, 91%). The observed focus on the structural attributes of compliant providers may be because these attributes were easier to measure and were available for quantitative analysis via routinely-collected data e.g. the Online Survey, Certification, And Reporting (OSCAR) database [185].

We find no other systematic review that has specifically focused on determinants of regulatory compliance in health and social care services. There are some reviews that used regulatory compliance as one of several outcome measures to evaluate associations between a specific structural or organisational characteristic (e.g. for-profit status [186] or service size [187] and quality). However, these studies were primarily interested in regulatory compliance as a proxy measure for quality while our review sought to identify structural or process determinants of compliance.

The processes that determined successful implementation of regulations (e.g. leadership engagement to ensure readiness for inspection) were much harder to map to routinely available data and were therefore harder to study. There were some CFIR constructs in the 'process' domain, which encompasses various stages of the implementation of an intervention such as 'planning', 'engaging–innovation participants' and 'reflecting & evaluating', to which we coded very few/no determinants. For example, we found no studies that investigated how organisations engaged in planning prior to the introduction of regulations.

We also coded few determinants to the 'characteristics of individuals' domain. Many of this domain's constructs deal with the views, opinions and perceptions of individuals towards an intervention. Such constructs, by their nature, lend themselves to qualitative research in contrast to routinely available quantitative data. Therefore, our review reveals a clear gap in the evidence base and opportunities for future qualitative or mixed-methods studies on successful engagement with the regulatory process.

## Facility characteristics

Our finding on facility size is in agreement with a similar review which investigated NH size and its relationship to quality [187]. Some have posited that staff attention and resources in larger facilities are directed towards health and quality of care outcomes at the expense of quality of life outcomes [188]. Larger facilities may also perform poorly with regard to broad quality measures (i.e. regulatory compliance) but better on individual quality indicators [187, 189]. Size may also interact with other facility characteristics that may impact on quality e.g. larger hospitals tend to be located in urban areas [190]. Facility size was also associated with staff turnover, where larger NHs have higher turnover, which may impact on quality [191]. We also found that high staff turnover was associated with compliance (see below).

Other studies have shown that for-profit facilities were associated with poorer compliance [186]; our findings are that 55% of studies reported a negative association. One explanation for the poorer performance may be that expenditures are minimised thereby resulting in fewer resources being directed towards care activities, which could in turn impact on compliance [186]. Not-for-profit facilities may be in a position to 'cherry-pick' clients (preferring people with minimal care needs and/or greater resources) whereas for-profit facilities may be more reliant on low-income clients with greater needs, thereby impacting on quality outcomes [74]. Higher process quality (e.g. leadership, training) in not-for-profit facilities may also be a mechanism through which higher quality is achieved [103].

Health service chains (mostly in the NH sector) have been the subject of considerable research with respect to quality, generally reporting a negative association [192–194]. We did not replicate this finding: 45% of studies in our review also reported a negative association but 26% reported a positive association and 29% a null association. Chain ownership accounts for varying proportions of the market across jurisdictions (e.g. in 2014, 64% of NHs were owned/operated by chains in the United Kingdom whereas the figure was 17% in Sweden and Canada) [192]. One possible explanatory mechanism is the association between chain status and lower rates of nurse staffing [16, 195], potentially impacting on a facility's compliance

**Table 7. Determinants of regulatory compliance where evidence is most consistent.**

| Domain | Construct | Determinant | Associated[*] | Negative | Null | Positive |
|---|---|---|---|---|---|---|
| Inner setting | Other | Higher staff turnover | | 16 | 3 | 0 |
| Inner setting | Structural characteristics | Geographic location of facility | 13 | | 1 | |
| Inner setting | Structural characteristics | Larger size facility (bed number) | | 40 | 16 | 3 |
| Inner setting | Other | Higher total nursing staff | | 2 | 3 | 11 |
| Inner setting | Other | Higher registered nurse staffing | | 1 | 9 | 19 |

* Denotes a nominal variable where a significant association was reported for one or more of those variables (e.g. federal states in Australia)

capability. The pursuit of corporate interests may also diminish resources [196], thereby affecting compliance. Some have cautioned that the literature on chain facilities tends to treat all chains equally, whereas there may be important differences in terms of chain size and ownership [196].

The association between occupancy levels and quality is also a facility characteristic subject to much research [197–200]. High occupancy levels (typically regarded as >90%) and overcrowding in hospitals have been associated with higher mortality [198, 201] and higher incidence of healthcare associated infections [202]. We found no clear association between occupancy and compliance. This is surprising as lower occupancy has been associated with poorer financial performance in nursing homes [203] and higher use of antipsychotic medication [200], which one might expect to also impact compliance. Given most studies in our review were set in NHs, it may be that NH compliance levels are less sensitive to occupancy compared to hospitals.

## Staffing

We find high staff turnover to be strongly associated with poorer compliance. A circular relationship may be evident here: poor quality care precipitates staff dissatisfaction and vice versa [191]. The high costs related to staff turnover [204] may impact on an organisation's compliance levels. High turnover rates have also been associated with poor implementation of evidence-based practices [205], thereby providing another potential causal mechanism for poorer compliance.

There was robust evidence that higher nurse staffing levels are associated with higher compliance. This finding is concordant with previous studies investigating nurse staffing levels and quality in hospitals [206–209] and NHs [200, 210, 211], albeit that some report inconclusive findings [212]. Some studies draw a distinction with respect to nurse type, finding registered nurses had a greater impact on quality when compared with licensed vocational nurses [200, 207, 208], a finding replicated in our review. Higher nurse staffing levels may be associated with quality when measures were nurse-sensitive outcomes (e.g. cardiac arrest, unplanned extubation) [206]. Lower nurse staffing levels may result in frontline care being provided by less-qualified staff which impacts on quality [208]. Higher levels of registered nurse staffing have been associated with lower antipsychotic medication use in NHs [200], which is also often a quality measure subject to regulation in these settings [213, 214]. Potential interaction effects with other staff-related measures (e.g. turnover, agency staff) have led some to argue that simply measuring nurse staffing levels is insufficient in terms of explaining the relationship to quality [215].

## Demographics

We find that the geographic location of a facility was strongly associated with compliance. Most studies that investigated this were interested in multiple other factors and were usually at national-level; the geographic location may have been different cities or states in America or Australia (as distinct from urban/rural or population density). It is likely that location was interacting with multiple other variables producing this association. One obvious explanation is inconsistent regulation, for which there is evidence across a range of industries [216–219]. The level of public funding available across regions can also differ substantially, thereby impacting the resources available for compliance [220, 221]. Geographic location is associated with health service availability [222], medically underserved populations [223] and NH use of antipsychotic medication [200], each of which may have implications for compliance.

Our findings indicate no clear association of service-users with more complex or resource-intensive needs, or populations with a more challenging profile (e.g. older, sicker or poorer) and compliance. This is somewhat surprising as there is evidence to suggest these populations have generally poorer access to quality healthcare. For example, facilities in severely-deprived neighbourhoods have lower staffing levels [224]. Moreover, greater resourcing is required for service users with higher acuity [225], which one might argue limits the resources available for compliance.

We found some evidence relevant to the processes by which implementation of regulations becomes 'normalised' within organisations. For example, the degree to which regulatory requirements were compatible with existing work practices was important [18]. Some studies in our review reported that habitually producing good documentation to substantiate regulatory compliance was also important (albeit that this was geared more towards 'gaming the system' as opposed to achieving the goals of regulation) [18, 31]. The above findings were echoed in studies relying on normalisation process theory (NPT) to evaluate implementation efforts. For example, in a study using NPT to evaluate the implementation of a regulatory intervention for individual doctors, the authors discussed how successful implementation was influenced by compatibility with existing practices and good documentation/record-keeping [226]. The process of interaction between regulator and regulatee also appears to influence the compliance outcome [33, 41, 137].

There was a very limited literature investigating behaviours of regulatory staff and their responses to varying levels of compliance. However, there was evidence that regulation was sometimes implemented poorly by regulators, thereby giving a false measure of compliance. For example, by inspectors effectively turning a blind eye [32], services finding workarounds [18], or the entire system of regulation representing a performative means by which to draw a veil over poor care practices [31]. There are two relevant issues here. Firstly, there are multiple examples in a range of industries where a public scandal erupts, implicating the regulator [227–229]. Poor implementation of regulations by a regulator (through mechanisms such as regulatory capture or incompetence) can have potentially serious consequences for the public good. Secondly, the circumvention of regulatory prescriptions (e.g. by workarounds or misleading documentation) is similar in nature to 'ritualism' [17]. Here, organisations engage in a subtle rejection of regulatory goals by 'playing the game' typified in the following example: "an Australian director of nursing. . .did not want to oppose an inspection team who 'made a big heap out of ethnic diet' under an Australian standard that requires sensitivity to cultural preferences for different types of food: 'So we bought ethnic diet books. . .give it a foreign name and they'll be happy'" [17]. This is compliance in name only and can mean that regulatory goals such as safety or person-centred care are not met.

There is an extensive literature on compliance theory which seeks to explicate compliance at the level of individuals, organisations and nation states across various regulated contexts such as tax [230, 231], transport [232, 233] and environmental protection [234, 235]. There are four conceptual themes that seek to explain compliance: "motives, organizational capacities and characteristics, regulation and enforcement, and social and economic environments (or institutions)" [236]. Motives include the pursuit of material (e.g. profit or limiting costs), emotional (fear of admonishment, anger) or normative (to act appropriately, do the right thing) goals [12]. Organisational capacity refers to the differences between organisations in terms of economic resources, knowledge and expertise [236]. Regulation and enforcement draws attention to the various strategies and styles adopted by regulators and how these influence compliance behaviour [236]. Finally, compliance can be affected by the wider environment (e.g. political, economic or social) such as when a regulator leverages public pressure through 'naming and shaming' [237].

There are two broad empirical approaches evident in the literature on compliance theory: objectivist and interpretive [236]. Objectivist efforts at explaining compliance test theories by regarding compliance as the dependent variable and seeking associations with specific independent variables. The interpretive school is more concerned with how compliance is socially constructed and contested: "The research task shifts from mapping 'compliance' and 'noncompliance'. . .to describing and understanding a whole range of organizational perceptions of, and behavioural responses to, regulation" [236].

Within the literature on compliance theory (which we searched separately) we found very few studies that sought to describe determinants of compliance in organisations within whole industries (e.g. banking or manufacturing). Rather, studies tended to focus on individual organisations [238] or were ethnographies that described how organisations/people engaged with regulation [239]. One review of financial regulation sought to identify organisational determinants of compliance [240]. The only statistically significant finding related to ownership structure: firms with a high separation of ownership from control (the degree to which board and shareholder power are separated) and family-owned firms were associated with higher compliance [240].

The lack of wide-ranging reviews make it difficult to situate our study within the wider literature on compliance theory. Some reflection on our findings in the light of the four conceptual themes and two empirical approaches outlined above may be useful. The studies in our review were located mostly within the organisational capacities and characteristics conceptual theme. These were largely objectivist in nature and identified variables that may be associated with compliance (e.g. staff levels, local demographics). There were only small numbers of studies that could be described as relevant for each of the other themes: motives; regulation and enforcement; and environment. Thus, the interpretive approach (which often relies on qualitative means of enquiry [236]) was largely absent from our review, representing a significant gap in the literature.

Our findings, particularly those that are supported by findings of other reviews, have important implications. Several modifiable organisational characteristics such as facility size and nurse staffing levels merit attention by policy makers, particularly in the context of NH design.

The inequity highlighted in the finding of poorer compliance for service users with higher needs and less resources (although not reaching our >60% threshold) is another reminder of the importance of the socio-economic determinants of health [241]. This represents a real challenge for regulators because while these populations deserve parity in terms of care quality, regulatory sanctions could potentially further diminish quality by diverting time/resources

away from care. Regulators may need to be cognisant of this issue, especially during times of economic uncertainty and pressure on public spending [242].

The design of regulatory interventions and their subsequent implementation could be informed by the application of implementation science. For example, regulatory impact analysis—an increasingly common process undertaken prior to the introduction of regulation [243]—could potentially benefit from identifying implementation barriers or facilitators.

There was an expectation on behalf of the authors that a larger number of qualitative studies would be found. There is an extensive field of qualitative enquiry on the implementation of other aspects of quality in health and social care such as clinical guidelines [244, 245], quality improvement initiatives [246, 247], patient feedback [248, 249], and professional regulation [250]. Moreover, the regulation and compliance literature is replete with qualitative studies of organisational responses to regulation, yet very few examine organisations working in health and social care [236]. The relative absence of studies on the implementation of regulatory standards and how this is linked to compliance is therefore puzzling and represents a gap in the literature. It may be the case that regulators are seen as a "remote disembodied agent" due to infrequent contact with managers and frontline staff, therefore being overlooked as a potential field of enquiry in implementation science [251].

Multiple avenues for future research arise from our findings. Firstly, the causal mechanisms for some determinants of compliance warrant deeper investigation across a range of contexts and settings. For example, while we find that larger facilities were associated with poorer compliance, some studies have found a positive association of quality and size [200, 252]. It may be that associations between facility size and quality differ depending on the quality measure assessed and further research may be able to address this matter.

Secondly, the 'regulatory character' of one setting/context can differ significantly to another: "findings from any one study need to be assessed in light of the particular jurisdiction and regulatory regime in question" [253]. Therefore, given the extensive body of research from the United States and in NHs, research focused on other countries and other settings could prove instructive.

Thirdly, the heavy focus on objectivist approaches could be complemented by more interpretive and qualitative research. For example, one may wish to understand how the behaviour of organisations (and individuals within those organisations) during regulatory encounters impacts on the compliance outcome, building on contemporary work in other settings [254].

Finally, there may be merit in exploring the utility of various implementation strategies that could potentially address barriers to regulatory compliance. For example, certain Expert Recommendations for Implementing Change (ERIC) strategies have been identified as most appropriate for addressing specific CFIR constructs [255]. Consideration of various implementation strategies could also inform a regulator's approach to inspection and enforcement practices.

## Strengths and limitations

A strength of this review is the application of a systematic, wide-ranging search that offered the best opportunity of identifying relevant literature. A published protocol guided the process and provided methodological rigour [26]. The authors are not aware of any other systematic review which focused specifically on regulatory compliance in health and social care services. In addition, the conceptualisation of regulation as an intervention, thereby facilitating use of the CFIR and implementation science more generally, is novel.

The ability to draw generalisable conclusions in this review is limited by the preponderance of literature focusing on NHs in the United States. Other settings like hospitals, and other

jurisdictions, are relatively absent in the literature. A further weakness in the evidence base is the reliance on observational designs which means causal inferences have to be tentative. We also draw attention to the mixed quality of the included studies as a limitation. Many studies relied on self-reported measures which could not be objectively verified. For example, many studies involving nursing homes relied on staffing levels as reported in the OSCAR database, which has been shown to be inconsistent when compared with other data sources [256].

A further limitation of our review is the heterogeneity of regulations used across the included studies. The heterogeneity is found both among studies (e.g. where one may focus specifically on infection control and another may cover a broad basket of regulations) and also in the diversity of regulatory frameworks across jurisdictions. It is likely that determinants vary in their importance depending on the specific regulation in question and the nature (e.g. timeliness, use of sanctions) of the regulatory relationship. This limits our ability to generalise about the importance of different determinants of compliance.

We considered a number of framework options for structuring our review. While we considered other options such as Normalisation Process Theory [257] and the Practical Robust Implementation and Sustainability Model (PRISM) [258], we ultimately chose to use the CFIR as it provided a comprehensive framework that encompassed both organisational and individual-level implementation features.

One might question whether regulation was a good fit for study under implementation science and the CFIR. Regulation can hardly be regarded as an 'innovation'—it is not new. Moreover, the 'ongoing' nature of regulation makes it different to other, more common, interventions which are perhaps time-limited [259, 260] or intended to be introduced and then subsumed (normalised) by an organisation [261, 262]. We encountered some coding difficulties with the CFIR, which perhaps reflects that initial implementation of an intervention and continuous quality assurance (i.e. regulation) are distinct. For example, organisations are required to implement and sustain regulatory standards on an ongoing basis, and have that monitored by a third-party (the regulator) at often random intervals. Such an intervention feature (i.e. sustainability and external evaluation) has no natural home within the CFIR. Nevertheless, we were satisfied that the CFIR provided a useful framework within which to evaluate regulation as an intervention.

There has been an acknowledgment that the CFIR has a blind spot in terms of implementation sustainability [263], indeed a revision to CFIR was recently published [264]. Discussions on CFIR utilisation in various contexts have also highlighted a need to differentiate between innovation and implementation outcomes [263]. In this review we regarded compliance as an implementation outcome: i.e. the degree to which organisations successfully implemented regulations. Similarly, we regarded determinants as constituting implementation determinants which predicted or explained actual implementation outcomes. Authors using the CFIR are encouraged to reflect on three questions which we have addressed in S1 Text.

## Conclusion

This review sought to identify and describe determinants of regulatory compliance. The multiplicity of determinants identified illustrates how regulatory compliance is a complex and context-specific phenomenon. There were some structural characteristics of organisations that showed an association with compliance. It is difficult to draw conclusive causal relationships and further research should attempt to better understand these mechanisms.

Regarding regulation as an intervention and using the CFIR as a framework offered some useful insights into the structures and processes that produce compliance. The uneven

distribution of determinants across the five CFIR domains demonstrates that the literature included in this review tends to draw on objectivist approaches.

There were only a small number of studies in this review that were qualitative and could be described as using an interpretive approach to understand compliance determinants. While such studies exist in other fields of regulation and compliance research, they are relatively absent in the context of health and social care and represent a gap in the literature.

## Supporting information

**S1 Checklist. PRISMA 2020 checklist.**
(DOCX)

**S1 Table. Search terms used on electronic databases.**
(DOCX)

**S2 Table. Full list of included studies.**
(DOCX)

**S1 File. Quality assessment of included studies.**
(XLSX)

**S1 Text. Author reflections on using the CFIR.**
(DOCX)

**S1 Dataset. Full dataset used for analysis.**
(XLSX)

## Acknowledgments

The authors wish to acknowledge the valuable contribution of Fiona Barry, University College Cork, Ireland in contributing to the quality assessment component of this systematic review.

## Author Contributions

**Conceptualization:** Paul Dunbar, Laura M. Keyes, John P. Browne.

**Data curation:** Paul Dunbar, Laura M. Keyes.

**Formal analysis:** Paul Dunbar, John P. Browne.

**Investigation:** Paul Dunbar.

**Methodology:** Paul Dunbar, Laura M. Keyes, John P. Browne.

**Project administration:** Paul Dunbar.

**Resources:** Paul Dunbar.

**Software:** Paul Dunbar.

**Supervision:** Laura M. Keyes, John P. Browne.

**Validation:** Paul Dunbar.

**Visualization:** Paul Dunbar.

**Writing – original draft:** Paul Dunbar, Laura M. Keyes, John P. Browne.

**Writing – review & editing:** Paul Dunbar, Laura M. Keyes, John P. Browne.

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
