## [Decision Letter · Decision Letter 0]

17 Jan 2023

PONE-D-22-30687Determinants of regulatory compliance in health and social care services: a systematic review using the Consolidated Framework for Implementation Research.PLOS ONE

Dear Dr. Dunbar,

Thank you for submitting your manuscript to PLOS ONE. After careful consideration, we feel that it has merit but does not fully meet PLOS ONE’s publication criteria as it currently stands. Therefore, we invite you to submit a revised version of the manuscript that addresses the points raised during the review process.

We look forward to receiving your revised manuscript.

Kind regards,

Ernesto Iadanza

Academic Editor

PLOS ONE

Journal Requirements:

 “This review comprises part of a PhD study undertaken by PD and funded by the Health Information and Quality Authority (Ireland). This research was conducted as part of the Structured Population health, Policy and Health-services Research Education (SPHeRE) programme (Grant No. SPHeRE/2019/1).

www.hiqa.ie

www.hrb.ie

https://www.sphereprogramme.ie”

4. Please amend the manuscript submission data (via Edit Submission) to include author Dr Laura Keyes and Prof John P Browne.

Reviewers' comments:

Reviewer's Responses to Questions

**Comments to the Author**

1. Is the manuscript technically sound, and do the data support the conclusions?

Reviewer #1: Yes

Reviewer #2: Yes

2. Has the statistical analysis been performed appropriately and rigorously? 

Reviewer #1: Yes

Reviewer #2: N/A

3. Have the authors made all data underlying the findings in their manuscript fully available?

Reviewer #1: Yes

Reviewer #2: Yes

4. Is the manuscript presented in an intelligible fashion and written in standard English?

Reviewer #1: Yes

Reviewer #2: Yes

5. Review Comments to the Author

Reviewer #1: This systematic review seeks to identify determinants of compliance with regulation in health and social care services. This is a well-executed and comprehensive systematic review. The results were synthesized in a narrative review using the constructs of the CFIR as grouping themes. Below are some suggestions for the authors to consider that may strengthen the manuscript.

1. One major concern is the variety of regulations in health and social care services. The heterogeneity of the included articles may be high. How can the authors draw conclusions from such different regulations? Were the data analysis reliable?

Abstract:

2. The CFIR was used as a framework for synthesizing the results. However, in the results section of abstract, the results were not presented based on such framework. Readers may not have an idea how the results were related to the framework by looking at the abstract.

Introduction:

3. It is better to add a paragraph about what previous studies have done to investigate the determinants of regulatory compliance in health and social care services. What were their findings? What frameworks or theories were used in their studies? Did any studies try to synthesize the findings of previous studies?

4. Please articulate the knowledge gap clearly near the end of the introduction.

5. The authors should give justifications for using the CFIR framework in this study. Why did the authors choose the CFIR, but not others (e.g., the Theoretical Domains Framework).

6. The authors used the CFIR as the framework for synthesizing the results. It is better to mention this approach was successfully used in previous studies. One article using such approach was listed below. Please cite it in the paper where appropriate.

Chan, P.Sf., Fang, Y., Wong, M.Cs. et al. Using Consolidated Framework for Implementation Research to investigate facilitators and barriers of implementing alcohol screening and brief intervention among primary care health professionals: a systematic review. Implementation Sci 16, 99 (2021). https://doi.org/10.1186/s13012-021-01170-8

Methods:

7. It is not clear about the inclusion and exclusion criteria of the studies. Please give more information.

8. For data analysis, this is a very important part. The authors should give details how the data coding of the data was performed. Any detailed approaches or procedures employed? Since the definitions of the CFIR constructs would be very different from the determinants of the study results, how were they matched? Given the current form, it is not clear about the data coding process.

Results:

9. I would suggest creating two tables for the results. Table 1 should summarize the findings of each study and code each determinant to the CFIR constructs. For Table 2, the findings should be presented according to the CFIR framework. As mentioned earlier, there was one publication of using the CFIR framework. Please look at the tables in that paper. By doing so, readers could get a better idea of what have been found in each of the previous studies and how the findings were coded under each CFIR construct. This suggestion is for the authors to improve the manuscript.

Chan, P.Sf., Fang, Y., Wong, M.Cs. et al. Using Consolidated Framework for Implementation Research to investigate facilitators and barriers of implementing alcohol screening and brief intervention among primary care health professionals: a systematic review. Implementation Sci 16, 99 (2021). https://doi.org/10.1186/s13012-021-01170-8

Discussion:

10. It is expected that there should have some discussions about implementation strategies that address the determinants in order to improve regulatory compliance. There are a host of implementation strategies. The authors should consider adding this section to the paper.

Look forward to reading the revised version of this interesting paper.

Reviewer #2: Review

Thank you for submitting this systematic review which adds to the literature on factors affecting compliance. The article which is interesting, detailed and well written and has a published protocol.

I have included some minor suggestions below

Only 20 of the existing 39 CIFR constructs were used. There is excessive used of an ‘other’ category Some of the determinants mentioned need to be moved to the relevant constructs.

Abstract

The writing needs to be consistent. A mixture of third person and first person styles are used e.g.

Results: The search yielded.. We found……

Introduction

Methods

Line 101. Needs to include a reference to regulatory compliance rather than just ‘compliance’

Analysis

Line 141. Explain what is meant by ‘different’ for nominal variables.

Results:

It would be helpful to indicate the number of constructs within each of the 5 headings (It is clear for inner setting) so that it is evident how may constructs out of the total are used in each domain.

Table 1.

Suggest to regroup European countries together in alphabetical order under ‘Country’ as n=1 in all cases.

Complexity Lines 265-6. Brief detail on the type of regulatory measure referred to here would help to explain why it would need its own QA meeting.

Line 285-286. The external influences referred to here are not just those that related to the needs and recourses of service users.

External policy and incentives includes funding. The point under ‘Other’ which relates to funding, Lines 351-353 should be moved here.

Lines 354-358 are External influences and should be moved to that section

Discussion

Some discussion on the mixed quality of the included studies is needed.

Reference to the updated version of CIFR Damschroder et al. Implementation Science (2022) 17:75https://doi.org/10.1186/s13012-022-01245-0 needs to be included.

6. PLOS authors have the option to publish the peer review history of their article (what does this mean?). If published, this will include your full peer review and any attached files.

Reviewer #1: No

Reviewer #2: **Yes: **Catherine Hayes

<quillbot-extension-portal></quillbot-extension-portal>

---

## [Author Response · Author response to Decision Letter 0]

1 Feb 2023

Response to academic editor

1. We have reviewed the PLOS One style requirements and have made amendments where appropriate. These were primarily in the following areas:

a. In-text citations changed from superscript in parentheses to normal text and square brackets. The positioning of the citations has also been amended in the manuscript (e.g. citations now appear before full-stops). I chose not to include these changes in the ‘Revised Manuscript with Track Changes’ as it would have made readability difficult for the reviewers.

b. Headings 1, 2 and 3 changed in accordance with style requirements. 

c. References to figures and tables have been amended in accordance with style requirements

2. The full dataset used for the analysis was included as supporting information with the previous submission, (Supplementary File 7). I can retain this here or can have it hosted online with a corresponding DOI, or both. 

3. I have amended the text in the manuscript to include the suggested line on Role of Funder. This change is also reflected in the cover letter.

4. I have amended the submission to include the names of both Prof John Browne and Dr Laura M Keyes. 

5. I have included captions in the ‘Supporting information’ section as requested and also made the necessary changes to the in-text citations. 

Response to Reviewer 1

[Reviewer comments in italics, response in normal text]

1. One major concern is the variety of regulations in health and social care services. The heterogeneity of the included articles may be high. How can the authors draw conclusions from such different regulations? Were the data analysis reliable?

We agree with the reviewer that this is a limitation of the study and should be stated more explicitly in the manuscript. As such, we have made the following addition to the text:

“A further limitation of our review is the heterogeneity of regulations used across the included studies. The heterogeneity is found both among studies (e.g. where one may focus specifically on infection control and another may cover a broad basket of regulations) and also in the diversity of regulatory frameworks across jurisdictions. It is likely that determinants vary in their importance depending on the specific regulation in question and the nature (e.g. timeliness, use of sanctions) of the regulatory relationship. This limits our ability to generalise about the importance of different determinants of compliance.”

2. The CFIR was used as a framework for synthesizing the results. However, in the results section of abstract, the results were not presented based on such framework. Readers may not have an idea how the results were related to the framework by looking at the abstract.

We agree that the abstract would be improved by making reference to the CFIR domain/constructs. We have amended the results section of the abstract which now reads as follows: 

“The search yielded 7,500 articles for screening, of which 157 were included. Most studies were quantitative designs in nursing home settings and were conducted in the United States. Determinants were largely structural in nature and allocated most frequently to the inner and outer setting domains of the CFIR. The following structural characteristics and compliance were found to be positively associated: smaller facilities (measured by bed capacity); higher nurse-staffing levels; and lower staff turnover. A facility’s geographic location and compliance was also associated. It was difficult to make findings in respect of process determinants as qualitative studies were sparse, limiting investigation of the processes underlying regulatory compliance.”

3. It is better to add a paragraph about what previous studies have done to investigate the determinants of regulatory compliance in health and social care services. What were their findings? What frameworks or theories were used in their studies? Did any studies try to synthesize the findings of previous studies?

We concur that this would be a useful addition to the manuscript. We have added the following text to the discussion section:

“We find no other systematic review that has specifically focused on determinants of regulatory compliance in health and social care services. There are some reviews that used regulatory compliance as one of several outcome measures to evaluate associations between a specific structural or organisational characteristic (e.g. for-profit status [184] or service size [185]) and quality. However, these studies were primarily interested in regulatory compliance as a proxy measure for quality while our review sought to identify structural or process determinants of compliance.” 

4. Please articulate the knowledge gap clearly near the end of the introduction.

We thank the reviewer for highlighting the need to more explicitly state the knowledge gap at the end of the introduction. We have now amended to relevant paragraph at the end of the introduction as follows:

“There is a body of work critiquing the effectiveness of regulation (e.g. improving system performance[22]), but given that regulation exists, and that there is much effort put into compliance, it is legitimate to investigate this practice on its own terms. There has been no previous synthesis of the extensive literature which makes use of regulatory compliance as an outcome measure in health and social care settings. Moreover, little is known about the processes by which health and social care services manage regulatory encounters and how this impacts on compliance. Thus, the aim of this systematic review is to identify and describe determinants of regulatory compliance in health and social care services, and the broader phenomenon of attempting compliance. The focus of the review is on the regulation of organisations as there are existing evidence syntheses on the regulation of individual professionals.[23, 24] “

5. The authors should give justifications for using the CFIR framework in this study. Why did the authors choose the CFIR, but not others (e.g., the Theoretical Domains Framework).

We did not consider using the Theoretical Domains Framework as we felt that it was focussed on the behaviour of individuals as opposed to organisations. Notwithstanding this, we acknowledge that there is no rationale offered for the choice of the CFIR in the manuscript. We are happy to remedy this and have now included the following text in the discussion section:

“We considered a number of framework options for structuring our review. While we considered using Normalisation Process Theory [253] and the Practical Robust Implementation and Sustainability Model (PRISM) [254], we ultimately chose to use the CFIR as it provided a comprehensive framework that encompassed both organisational and individual-level implementation features.” 

6. The authors used the CFIR as the framework for synthesizing the results. It is better to mention this approach was successfully used in previous studies. One article using such approach was listed below. Please cite it in the paper where appropriate.

Chan, P.Sf., Fang, Y., Wong, M.Cs. et al. Using Consolidated Framework for Implementation Research to investigate facilitators and barriers of implementing alcohol screening and brief intervention among primary care health professionals: a systematic review. Implementation Sci 16, 99 (2021). https://doi.org/10.1186/s13012-021-01170-8

We thank the reviewer for this suggestion and agree that it supports our decision to use the CFIR to structure our synthesis. We have included the following sentence and reference in the introduction:

“The CFIR has been used previously as a framework for structuring and synthesising the results of systematic reviews [22].”

7. It is not clear about the inclusion and exclusion criteria of the studies. Please give more information.

The reviewer will note that we direct the reader to the published protocol for the review and state that the methods in the current paper are merely summarised. This was done in an effort to limit the word count in the manuscript. However, we agree that it would be useful to include this important aspect of the methods in the manuscript and have included the text below accordingly: 

Criteria for inclusion

The phenomena of interest were determinants of regulatory compliance in health and social care services.

There were no limits on the articles for inclusion in terms of publication date or language.

Articles — either qualitative, quantitative or mixed-methods — were included if they:

• Described factors or characteristics that were related to regulatory compliance. Specifically, this refers to regulations that are mandated by government or other state authorities. A wide range of constructs were considered for inclusion including, but not limited to, the following: service characteristics (size, location, model of care, ownership); organisational characteristics (culture, management/governance structure, maturity); service user characteristics (age, disability type, disease/illness); nature of engagement (punitive, adversarial, collaborative).

• Discussed barriers or facilitators to regulatory compliance for health and social care services.

• Were focused on quality of care in health and social care services and used regulatory compliance as an outcome measure.

Studies were excluded if they:

• Analysed regulatory compliance in a field other than in a health or social care setting or service.

• Analysed compliance with clinical guidelines or other evidence-based methods for managing care that were not underpinned by the potential for regulatory sanction where there was a failure to comply.

• Used an outcome measure that was not equivalent to regulatory compliance in accordance with the definitions set out above. For example: adherence to voluntary standards or codes of conduct; where failure to comply does not result in regulatory sanctions of enforcement; compliance concerning individuals as opposed to organisations as is the case with regulations for specific health care professionals.

8. For data analysis, this is a very important part. The authors should give details how the data coding of the data was performed. Any detailed approaches or procedures employed? Since the definitions of the CFIR constructs would be very different from the determinants of the study results, how were they matched? Given the current form, it is not clear about the data coding process.

We thank the reviewer for this suggestion and are happy to provide further detail in the manuscript on the coding process. We have amended the relevant section in the methods as follows:

“Determinants were then coded to the most appropriate CFIR construct through an iterative and deliberative process among all three authors. PD first coded each determinant to the most appropriate CFIR construct (e.g. determinants related to facility size were coded to the structural characteristics construct in the inner setting domain), with reference to established definitions [21]. These data were then discussed jointly by PD and LMK in order to ensure that determinants were placed in the most appropriate construct. This process resulted in some determinants being moved to alternative constructs, and also the creation of a category of ‘other’ within each domain. This was necessary due to the significant number of determinants that had no appropriate match in the CFIR (e.g. the tools used by inspectors had no equivalent match for a construct in the intervention characteristics domain). 

PD, LMK and JB subsequently discussed all coding with a particular emphasis on whether the determinant had relevance in the specific context of implementing regulations. For example, a determinant relating to the level of public/state funding made available for services was originally coded to the available resources construct in the inner setting domain. On reflection, we decided that this was not appropriate as the funding was not specifically related to implementing regulations. As such, this determinant was moved to ‘other’ in the inner setting domain. This deliberation process continued until all three authors agreed on the coding of each determinant. The analysis was also consistent with the application of the CFIR to post-implementation evaluation of innovations, where determinants are linked to outcomes i.e. compliance [21]. “

9. I would suggest creating two tables for the results. Table 1 should summarize the findings of each study and code each determinant to the CFIR constructs. For Table 2, the findings should be presented according to the CFIR framework. As mentioned earlier, there was one publication of using the CFIR framework. Please look at the tables in that paper. By doing so, readers could get a better idea of what have been found in each of the previous studies and how the findings were coded under each CFIR construct. This suggestion is for the authors to improve the manuscript.

Chan, P.Sf., Fang, Y., Wong, M.Cs. et al. Using Consolidated Framework for Implementation Research to investigate facilitators and barriers of implementing alcohol screening and brief intervention among primary care health professionals: a systematic review. Implementation Sci 16, 99 (2021). https://doi.org/10.1186/s13012-021-01170-8

The suggestion to include additional tables is helpful and we agree that it would improve the manuscript. Therefore, we have opted to include five additional tables, at the start of each CFIR domain in the results, outlining the determinants found for each construct, along with the references of each study that investigated that determinant.

10. It is expected that there should have some discussions about implementation strategies that address the determinants in order to improve regulatory compliance. There are a host of implementation strategies. The authors should consider adding this section to the paper.

We concur with the reviewer’s suggestion that a discussion on implementation strategies would improve the manuscript. We have included the following paragraph in the discussion section:

“Finally, there may be merit in exploring the utility of various implementation strategies that could potentially address barriers to regulatory compliance. For example, certain Expert Recommendations for Implementing Change (ERIC) strategies have been identified as most appropriate for addressing specific CFIR constructs [254]. Consideration of various implementation strategies could also inform a regulator’s approach to inspection and enforcement practices. “

Response to Reviewer 2

[Reviewer comments in italics, response in normal text]

1. Only 20 of the existing 39 CIFR constructs were used. There is excessive used of an ‘other’ category Some of the determinants mentioned need to be moved to the relevant constructs.

The reviewer touches on an important aspect of how we coded determinants to CFIR constructs. The coding of determinants was an iterative and deliberative process that involved all three authors (please see amendment to methods described in point 8 above). We ultimately concluded that some determinants were only relevant to CFIR constructs if they were specifically related to the implementation of regulations. For example, while determinants related to funding may be considered appropriate for CFIR constructs such as cost or external policy and incentives, we only placed them here if the funding was expressly linked to an effort to implement regulations. General funding made available by the state was considered core to a service and not specific to the implementation of regulations. Another example is found in staffing levels. While one might argue that staffing levels could be coded to available resources, we concluded that this would only be appropriate in cases where the staff were explicitly engaged in implementing regulations or somehow addressing barriers to implementation. 

To further explain our decision, our aim in using the CFIR was not an attempt to fit all determinants to domains and constructs. Rather, it was a tool by which to structure and organise our analysis. As such, we were only prepared to code determinants to constructs where we were satisfied that it fit within the relevant definition and also made sense from an implementation perspective. 

2. The writing needs to be consistent. A mixture of third person and first person styles are used e.g.

Results: The search yielded.. We found……

We agree that this requires amending in the abstract. We have duly removed any reference to the first person.

3. Line 101. Needs to include a reference to regulatory compliance rather than just ‘compliance’

We concur that this requires revision. As we have previously provided a definition for compliance in the paper, we have changed this section of the text to make clear that it refers to the authors’ interpretation of regulatory compliance for the purposes of the analysis. New text reads as follows: “We interpreted regulatory compliance as any formal approval of the performance of a health or social care provider by the relevant regulator or accrediting agency.”

4. Line 141. Explain what is meant by ‘different’ for nominal variables.

We thank the reviewer for this suggestion as it will improve understanding of the manner in which variables were treated in the review. The text has been amended to include the following: “‘different/null’ for nominal variables (‘different’ denoting that one or more categories in the variable was positively or negatively associated with compliance).”

5. It would be helpful to indicate the number of constructs within each of the 5 headings (It is clear for inner setting) so that it is evident how may constructs out of the total are used in each domain.

We agree that this would be a useful edit to the manuscript and one which will help the reader. We have amended the text as follows: 

• “We coded nothing to four of the eight constructs in this domain: intervention source; trialability; design quality and packaging; cost.”

• “No determinants were coded to two of four constructs in this domain: ‘cosmopolitanism’ or ‘needs and resources of those served by the organisation’.”

• “We found no determinants that could be coded to two of the five constructs in this domain: knowledge and beliefs about the innovation, and individual stage of change.”

• “No studies were coded to six of the 10 constructs within the domain: planning; engaging – opinion leaders; engaging – formally appointed internal implementation leaders; engaging – champions; engaging – external change agents; engaging – innovation participants.”

6. Table 1. Suggest to regroup European countries together in alphabetical order under ‘Country’ as n=1 in all cases.

We have amended the table as suggest, see below:

Country n (%)

USA 140 (89.2)

Australia 9 (5.7)

Europe

 United Kingdom 3 (1.9)

 Denmark 1 (0.6)

 Portugal 1 (0.6)

 Netherlands 1 (0.6)

China 1 (0.6)

Saudi Arabia 1 (0.6)

7. Complexity Lines 265-6. Brief detail on the type of regulatory measure referred to here would help to explain why it would need its own QA meeting.

We agree that the edit suggested above would add important information for the reader and have amended the text as follows:

“A regulatory measure related to sepsis treatment procedures presented “unprecedented complexity”;”

8. Line 285-286. The external influences referred to here are not just those that related to the needs and recourses of service users.

Having reviewed the lines referred to above it is unclear to us what aspect of the text is at issue. The lines contain a quotation from Safaeinili et al (2020) and it is used as a means of defining the outer setting domain. Perhaps there is some confusion here as we do not see how the text implies that external influences are only related to the needs and resources of service users. 

9. External policy and incentives includes funding. The point under ‘Other’ which relates to funding, Lines 351-353 should be moved here.

We have addressed this point in number 1 above.

10. Some discussion on the mixed quality of the included studies is needed.

We agree with this suggestion that, in addition to the section on quality appraisal in the results, we should also make reference to this as a limitation in the discussion. We have added the text below to the strengths and limitations section:

“We also draw attention to the mixed quality of the included studies as a limitation. Many studies relied on self-reported measures which could not be objectively verified. For example, many studies involving nursing homes relied on staffing levels as reported in the OSCAR database, which has been shown to be inconsistent when compared with other data sources [255].”

11. Reference to the updated version of CIFR Damschroder et al. Implementation Science (2022) 17:75https://doi.org/10.1186/s13012-022-01245-0 needs to be included.

We thank the reviewer for this suggestion as this was an oversight on our behalf. We have no included a reference to the updated CFIR published in 2022 in the following section of text: 

“There has been an acknowledgment that the CFIR has a blind spot in terms of implementation sustainability [257], indeed a revision to CFIR was recently published [258].”

---

## [Decision Letter · Decision Letter 1]

13 Mar 2023

Determinants of regulatory compliance in health and social care services: a systematic review using the Consolidated Framework for Implementation Research.

PONE-D-22-30687R1

Dear Dr. Dunbar,

We’re pleased to inform you that your manuscript has been judged scientifically suitable for publication and will be formally accepted for publication once it meets all outstanding technical requirements. But. please, address the small minor revisions suggested by Reviewer 2.

Kind regards,

Ernesto Iadanza

Academic Editor

PLOS ONE

Additional Editor Comments (optional):

Reviewers' comments:

Reviewer's Responses to Questions

**Comments to the Author**

1. If the authors have adequately addressed your comments raised in a previous round of review and you feel that this manuscript is now acceptable for publication, you may indicate that here to bypass the “Comments to the Author” section, enter your conflict of interest statement in the “Confidential to Editor” section, and submit your "Accept" recommendation.

Reviewer #1: All comments have been addressed

Reviewer #2: (No Response)

2. Is the manuscript technically sound, and do the data support the conclusions?

Reviewer #1: Yes

Reviewer #2: Yes

3. Has the statistical analysis been performed appropriately and rigorously? 

Reviewer #1: Yes

Reviewer #2: N/A

4. Have the authors made all data underlying the findings in their manuscript fully available?

Reviewer #1: Yes

Reviewer #2: Yes

5. Is the manuscript presented in an intelligible fashion and written in standard English?

Reviewer #1: Yes

Reviewer #2: Yes

6. Review Comments to the Author

Reviewer #1: (No Response)

Reviewer #2: Dear Dr Dunbar,

You have satisfactorily addressed my comments and suggestions. However I noticed that on page 10 there is an insertion of a figure 1 heading in error.

Also, on Figure 1 the heading should be placed below the figure and not above.

Once these very small issues have been addressed, in my view the manuscript will be suitable for publication in Plos One.. Well done.

Catherine Hayes

Reviewer

7. PLOS authors have the option to publish the peer review history of their article (what does this mean?). If published, this will include your full peer review and any attached files.

Reviewer #1: No

Reviewer #2: **Yes: **Catherine B Hayes

---

## [Editor Report · Acceptance letter]

4 Apr 2023

PONE-D-22-30687R1 

Determinants of regulatory compliance in health and social care services: a systematic review using the Consolidated Framework for Implementation Research. 

Dear Dr. Dunbar:

I'm pleased to inform you that your manuscript has been deemed suitable for publication in PLOS ONE. Congratulations! Your manuscript is now with our production department. 

Kind regards, 

on behalf of

Dr. Ernesto Iadanza 

Academic Editor

PLOS ONE